## TOPICAL REVIEW

# Sodium–glucose cotransporter 2 inhibitor ameliorates high fat diet-induced hypothalamic–pituitary–ovarian axis disorders

Xiaolin Chen[1], Lili Huang[2], Ling Cui[3], Zhuoni Xiao[4], Xiaoxing Xiong[5] and Chen Chen[2] 

[1] *Department of Endocrinology, Renmin Hospital of Wuhan University, Wuhan, China*

[2] *School of Biomedical Science, University of Queensland, Brisbane, Queensland, Australia*

[3] *Department of Reproduction and Infertility, Chengdu Women's and Children's Central Hospital, School of Medicine, University of Electronic Science and Technology of China, Chengdu, China*

[4] *Reproductive Medical Center, Renmin Hospital of Wuhan University, Wuhan, China*

[5] *Department of Neurosurgery, Renmin Hospital of Wuhan University, Wuhan, China*

Handling Editors: Laura Bennet & Rebecca Simmons

The peer review history is available in the Supporting Information section of this article (https://doi.org/10.1113/JP283259#support-information-section).

**The Journal of Physiology**

**Abstract** High-fat diet (HFD) consumption is known to be associated with ovulatory disorders among women of reproductive age. Previous studies in animal models suggest that HFD-induced microglia activation contributes to hypothalamic inflammation. This causes the dysfunction of the hypothalamic–pituitary–ovarian (HPO) axis, leading to subfertility. Sodium–glucose

X. Chen and L. Huang contributed equally to this work.

cotransporter 2 (SGLT2) inhibitors are a novel class of lipid-soluble antidiabetic drugs that target primarily the early proximal tubules in kidney. Recent evidence revealed an additional expression site of SGLT2 in the central nervous system (CNS), indicating a promising role of SGLT2 inhibitors in the CNS. In type 2 diabetes patients and rodent models, SGLT2 inhibitors exhibit neuroprotective properties through reduction of oxidative stress, alleviation of cerebral atherosclerosis and suppression of microglia-induced neuroinflammation. Furthermore, clinical observations in patients with polycystic ovary syndrome (PCOS) demonstrated that SGLT2 inhibitors ameliorated patient anthropometric parameters, body composition and insulin resistance. Therefore, it is of importance to explore the central mechanism of SGLT2 inhibitors in the recovery of reproductive function in patients with PCOS and obesity. Here, we review the hypothalamic inflammatory mechanisms of HFD-induced microglial activation, with a focus on the clinical utility and possible mechanism of SGLT2 inhibitors in promoting reproductive fitness.

(Received 6 May 2022; accepted after revision 16 August 2022; first published online 1 September 2022)

**Corresponding author** C. Chen: School of Biomedical Sciences, MacGregor Building 409A, University of Queensland, Brisbane 4072, Australia. Email: chen.chen@uq.edu.au

X. Xiong: Department of Neurosurgery, Renmin Hospital of Wuhan University, Wuhan, 430060, China. Email: xiaoxingxiong@whu.edu.cn

**Abstract figure legend** Summary of high fat diet (HFD)-induced anovulation and amelioration of this dysfunction by sodium–glucose cotransporter 2 (SGLT2) inhibitors. Hypothalamic inflammation caused by HFD impairs gonadotropin-releasing hormone (GnRH) surge release and leads to ovulatory dysfunction, a progression mediated by proinflammatory factors secreted by activated microglia. In addition, the improvement of ovulatory dysfunction by SGLT2 inhibitors may be mediated by repair of GnRH surge release and neuroprotective properties. The pathways for neuroprotective properties of SGLT2 inhibitors are: (1) inhibition of microglial activation, resulting in decreased proinflammatory factor release and neuroinflammation; (2) alleviation of cerebral atherosclerosis; (3) reduction of oxidative stress in neurons; and (4) inhibition of reactive oxygen species (ROS)-dependent neuronal apoptosis.

## Introduction

The consumption of a high-fat diet (HFD) in humans and animals increases the risk of developing type 2 diabetes (T2D), cardiovascular diseases and cognitive impairment, and also has a negative impact on reproductive fitness in both males and females (Chakraborty et al., 2016; Cheng et al., 2019; Holloway et al., 2011; McLean et al., 2019; Skaznik-Wikiel et al., 2016). Energy homeostasis is a key regulator of the reproductive system. Excessive energy consumption is closely related to hormonal disturbances and reproductive disorders. Polycystic ovary syndrome (PCOS) is a common reproductive endocrine and metabolic diseases affecting 4–7% of reproductive-age women (Bozdag et al., 2016). Women with PCOS are more inclined to consume a HFD (Mizgier et al., 2021). Anovulation is the key characteristic of PCOS, which is often accompanied by insulin resistance, hyperandrogenism, hyperlipidaemia and abdominal obesity (Osibogun et al., 2020). Notwithstanding advancements in research regarding the mechanisms for such a tight relationship between energy excess and reproductive dysfunction, the underlying pathogenesis of PCOS has not been elucidated. Although various rodent models have deepened our understanding of PCOS aetiology (Patel & Shah, 2018; Roberts et al., 2017), there is as

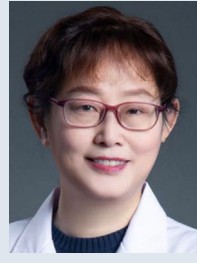
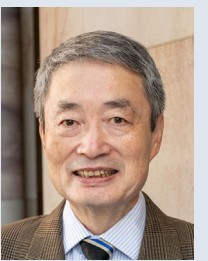

**Xiaolin Chen** is employed in the department of endocrinology in Renmin Hospital of Wuhan University as a doctor and vice professor. She trained at the University of Queensland's School of Biomedical Science before returning to Renmin Hospital of Wuhan University to investigate how excessive energy expenditure affects the function hypothalamic–pituitary–ovarian axis. **Chen Chen** is currently the Chair Professor of Endocrinology at the School of Biomedical Sciences, Faculty of Medicine, The University of Queensland. He has medical training from Fudan University (Shanghai) and PhD from University of Bordeaux, France. He has done extensive research in molecular and cellular neuroendocrinology and endocrinology of metabolic disorders.

yet no one-size-fits-all rodent model that exhibits all the clinical traits of PCOS. Previous studies in female mice have shown that HFDs cause increased body fat content, insulin resistance, irregular oestrous cycling and ovulatory dysfunction (Lai et al., 2014; Wu et al., 2019). Similarly, feeding female rats a HFD from pre-pubertal age produced both reproductive and metabolic disturbances mimicking the features of PCOS (Patel & Shah, 2018). Thus, the HFD-induced obesity mouse model has been widely used to study the underlying cause of PCOS. Although the exact pathophysiological mechanisms underlying the characteristics of the PCOS phenotype remain largely unknown, a commonly accepted concept is metabolic disturbance, which is often associated with insulin resistance. Therefore, it is not surprising that metformin is used as the first line therapeutic medicine (Lovvik et al., 2019). However, as emerging novel drugs have been evidenced to improve insulin sensitivity and energy homeostasis in diabetic patients, we are in position to reveal the potential of these drugs in treating PCOS.

Sodium–glucose cotransporter 2 (SGLT2) inhibitors are a novel group of anti-hyperglycaemia drugs for treating T2D. The main mechanism of their action is to block glucose at SGLT2 receptors in the proximal tubules of the kidney and inhibit urine glucose reabsorption. In addition to the direct effects of SGLT2 inhibitors in improving glucose homeostasis and insulin sensitivity in diabetes patients, many large double-blind clinical observations and a meta-analysis revealed additional benefits in cardiovascular mortality, stroke events and kidney protection in treatment of diabetic patients (Perkovic et al., 2019; Wiviott et al., 2019; Zinman et al., 2015). Considering that the key metabolically disturbed trait of PCOS (defined as insulin resistance) resembles that of diabetes, and there are common accompanying features of the two (including obesity, low grade inflammation, metabolic abnormalities and increased risk of cardiovascular events) (Berni et al., 2021; Wekker et al., 2020), it is reasonable to propose that SGLT2 inhibitors could be beneficial for patients with PCOS, and it is worthwhile exploring any emerging mechanisms that are supplementary except for improving metabolic abnormalities.

In this review, we summarize the mechanism of HFD-induced reproductive dysfunction, with a specific focus on hypothalamic inflammation and hypothalamic–pituitary–ovarian (HPO) axis disorder. Moreover, we outline the clinical and experimental studies of SGLT2 inhibitors in improving metabolic disorders and HPO axis dysfunction in PCOS patients or animal models. Finally, we aim to explore the possible mechanism of SGLT2 inhibitors in improving HPO axis dysfunction and provide a theoretical basis for further research in reproductive endocrine and metabolic diseases.

## Consumption of HFD causes hypothalamic–pituitary–ovarian axis dysfunction

Notwithstanding the limitations in fully representing the pathophysiological traits of PCOS in humans, over recent decades animal models have taken our understanding of the aetiology of PCOS beyond human studies. Among many animal models, the HFD-induced obese mouse model is the most widely used due to its abundant traits related to PCOS, including increased body fat content, development of insulin resistance, irregular oestrous cycling and ovulatory dysfunction (Negron & Radovick, 2020; Ziarniak et al., 2018). HFD-related reproductive abnormalities in female mice include early vaginal opening, irregular oestrous cycles, abnormal luteinizing hormone (LH) release and compromised fertility due to anovulation (Mikhael et al., 2019). It is believed that HFD exposure induces reproductive dysfunction at multiple levels of the HPO axis leading to ovulation disorders. Ovulation is triggered by a LH surge, which is secondary to a surge-mode release of gonadotropin-releasing hormone (GnRH) by GnRH neurons. Studies have evaluated the effects of a HFD on GnRH secretion and reproductive function (Brothers et al., 2010; Chang et al., 2021; Hussain et al., 2016; Ma et al., 2016) and revealed that the GnRH pulse generator is extremely sensitive to energetic stress. In one study, it was reported that HFD-induced obese female DBA/2J mice manifested a more than 50% suppression of GnRH expression accompanied by more than 60% decrease in natural pregnancy rates (Tortoriello et al., 2004) (Fig. 1). In another study, C57BL/6J female mice were placed on a 60% HFD for 32 weeks (Skaznik-Wikiel et al., 2016). Both obese mice and non-obese mice were in a state of subfertility and had impaired ovarian function regardless of the obese phenotype (Skaznik-Wikiel et al., 2016). This result suggests that a HFD itself may cause the female reproductive dysfunction independent of the progression of obesity. It is likely that the HFD acts on the reproductive neurons to alter their secretion patterns as the pulsatile release of GnRH and LH is critical for follicle maturation in female mammals. This is supported by a recent study showing that a HFD increased LH pulse frequency at dioestrus and reduced mean and basal LH levels at oestrus in mice (Negron & Radovick, 2020) (Fig. 1). Moreover, in rodents, evidence has shown that nitric oxide (NO) produced by NO-synthesizing neurons is required for kisspeptin-dependent oestrogen positive feedback and is responsible for ovulation (Bellefontaine et al., 2014; Constantin et al., 2021). Consumption of a HFD reduces the number of neuronal NOS-expressing neurons in the cortex (Gzielo et al., 2017). Therefore, studies are needed to test the hypothesis that HFD feeding reduces the kisspeptin-dependent activation of NO-synthesizing neurons in the anteroventral peri-

ventricular nuclei leading to inhibition of the surge release of GnRH.

In addition to the impact on central neurons, HFD also showed detrimental effects on ovary and oocyte quality. Compromised fertility and reduced primordial follicles in the ovary were observed in HFD mice, which was accompanied by high levels of proinflammatory factors and increased macrophage infiltration in the ovary (Gao et al., 2021; Hohos et al., 2020; Skaznik-Wikiel et al., 2016). The ovaries from HFD-induced DIO mice show more atretic follicles and contain mature follicles with fewer granulosa cells (Wen et al., 2020; Wu et al., 2017). The suggested mechanisms responsible for such ovarian dysfunction following long-term exposure to HFD include increased granulosa cell apoptosis and the suppression of ovarian angiogenesis. Granulosa cell apoptosis may be largely attributed to oxidative stress-associated DNA damage and end-oplasmic reticulum stress induced by the HFD (Wen et al., 2020; Wu et al., 2017) (Fig. 1). The suppression of ovarian angiogenesis, on the other hand, is considered a consequence of the HFD-induced inhibition of hypoxia inducible factor 1$\alpha$–vascular endothelial growth factor signalling, which subsequently arrests follicular development and formation of corpus luteum (Wu et al., 2017). In addition, other key ovarian genes are also dysregulated by a HFD. For example, Hohos et al. have found that expression of the endothelin-2 gene in ovary, a key gene for ovulation, is completely dysregulated during the oestrous cycle in HFD-fed mice. Simultaneously, key ovarian steroidogenic genes, such as *Star* and *Cyp19a1*, are also dysregulated after a HFD (Hohos et al., 2021).

Thus, a HFD causes HPO axis dysfunction traversing multiple levels, from the central regulating neurons down to the peripheral ovarian glands. Inflammation induced by a HFD is generally accepted as a key molecular mechanism at levels of peripheral organs. It is unclear whether a HFD directly contributes to hypothalamic inflammation that affects the major reproductive regulator neurons and leads to the dysfunction of the HPO axis.

## HFD-induced hypothalamic inflammation is involved in HPO axis dysfunction

**Microglia activation contributes to neuroinflammation.** Microglia are the resident macrophages of the brain and require colony-stimulating factor 1 receptor signalling for their survival (Alliot et al., 1999; Ginhoux et al., 2010). Representing 5–10% of the total cell population within the brain parenchyma (Aguzzi et al., 2013), microglia exert a major role in the immune response and preservation of tissue homeostasis by clearing dying cells, pathogens, aberrant proteins and debris (Faustino et al., 2011; Maas et al., 2020; Nugent et al., 2020; Villani et al., 2019). By responding to a wide variety of pathological stimuli, microglia scrutinize brain development, monitor synaptic function and regulate synaptogenesis to protect the central nervous system (CNS) from various pathological conditions throughout life (Hammond et al., 2019; Lund et al., 2018; Nguyen et al., 2020; Saitgareeva et al., 2020; Siew et al., 2019; Wake et al., 2009). Microglia in the CNS have been shown to be extremely plastic and can assume distinctive phenotypes including a homeostatic state and an activated state in response to various stimulations and depending on the local microenvironment (Kalambogias et al., 2020; Klawonn et al., 2021; Pasciuto et al., 2020). Single-cell and single-nucleus RNAseq have revealed that each subtype of microglia displays inherent properties and has distinctive functions (Clark et al., 2021; Colonna & Brioschi, 2020; Lloyd et al., 2019). The numerous functions of microglia are achieved through diverse phenotypes, each associated with a unique molecular signature (Song & Colonna, 2018). Although they are potentially multi-functional, microglia are generally

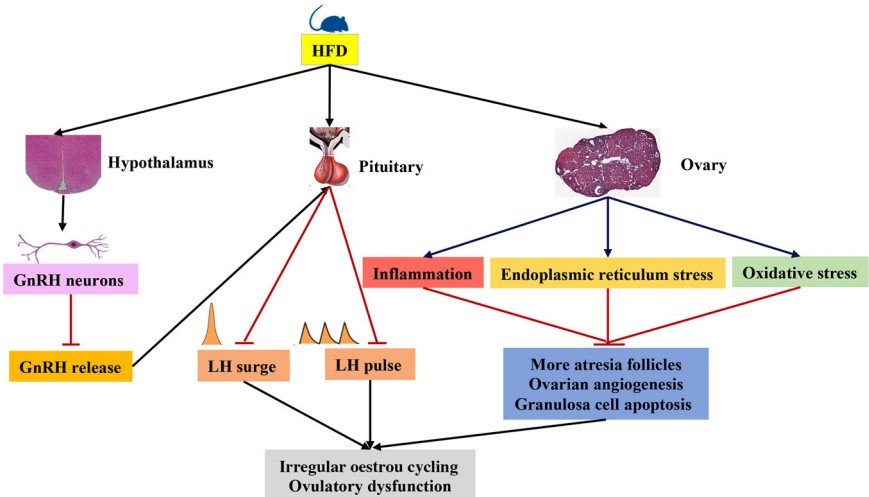

**Figure 1. HFD exposure induces hypothalamic–pituitary–ovarian (HPO) axis dysfunction leading to ovulation disorders**
HFD may act on hypothalamic GnRH neurons and anterior pituitary to alter the surge and pulsatile secretion of GnRH and LH, resulting in irregular oestrous cycling and ovulation disorders. HFD may also act on ovary, inducing inflammation, endoplasmic reticulum (ER) stress and oxidative stress, leading to increased atresia follicles, granulosa cell (GC) apoptosis and the suppression of ovarian angiogenesis. Black arrow, stimulation; red blunt line, inhibition.

thought to benefit neurons when activated under suitable physiological conditions. However, under pathological conditions, activated microglia release inflammatory factors to induce neuroinflammation, which triggers neuronal death (Alexaki et al., 2018; Marschallinger et al., 2020; Ndoja et al., 2020). All microglia express ionized calcium-binding adaptor molecule 1 (IBA1), and major histocompatibility complex II (MHCII) is considered a marker of activated microglia and is expressed on their surface (Hopperton et al., 2018). Proinflammatory microglia are characterized by amoeboid morphology and high expression of CD68, CD80 and CD11b (Merlini et al., 2019; Unger et al., 2018; Wachholz et al., 2016). The output of these proinflammatory microglia includes interleukin (IL)-1$\beta$, IL-6, IL-18, nitric oxide (NO), tumour necrosis factor-$\alpha$ (TNF-$\alpha$), reactive oxygen species (ROS) and proteases (Lively & Schlichter, 2018; Plastira et al., 2017). In addition, activated microglia are also involved in neuroprotection and repair. These microglia exhibit an anti-inflammatory action, high expression of IBA1, MHCII, arginase-1 and CD206, and release of IL-10, IL-4, and transforming growth factor $\beta$ (Orihuela et al., 2016; Plastira et al., 2016; Vergara et al., 2019) (Fig. 2).

Many neurodegenerative diseases such as Alzheimer disease (AD), Parkinson's disease (PD) and amyotrophic lateral sclerosis are linked with neuroinflammation induced by activated microglia (Hickman et al., 2018). Activation of the NOD-, LRR- and pyrin domain-containing protein 3 (NLRP3) inflammasome in microglia plays a key role in the development and progression of neuroinflammation. Ising et al. (2019) showed lower levels of tau hyperphosphorylation and IL-1$\beta$ in NLRP3 inflammasome-deficient mice. A20 is a nuclear factor $\kappa$B (NF-$\kappa$B) regulatory protein in microglia. Previous research has reported that mice with microglial A20 deficiency show hyperactivation of the NLRP3 inflammasome and neuroinflammation (Voet et al., 2018). The microglial NLRP3 inflammasome is a multimeric complex containing NLRP3, apoptosis-associated speck-like protein containing C-terminal caspase-activation and recruitment domain (CARD) (ASC), and pro-caspase 1. Upon activation, NLRP3 interaction with ASC through CARD–CARD results in recruitment of pro-caspase 1, which is then converted to caspase 1 (Mariathasan et al., 2004). Activated caspase 1 cleaves and activates pro-IL-1$\beta$ and pro-IL-18 into IL-1$\beta$ and IL-18, respectively. Finally, the activated microglia release IL-1$\beta$ and IL-18 contributing to development and progression of neuroinflammation and disease (Freeman et al., 2017; Voet et al., 2019) (Fig. 3)

**High-fat diet induces pathological hypothalamic microglia activation.** HFDs enriched with long chain saturated fatty acids (LCSFAs) are known to directly impact the CNS (D'Alonzo et al., 2020; Milanski et al., 2009; Valdearcos et al., 2014). Excessive intake of dietary

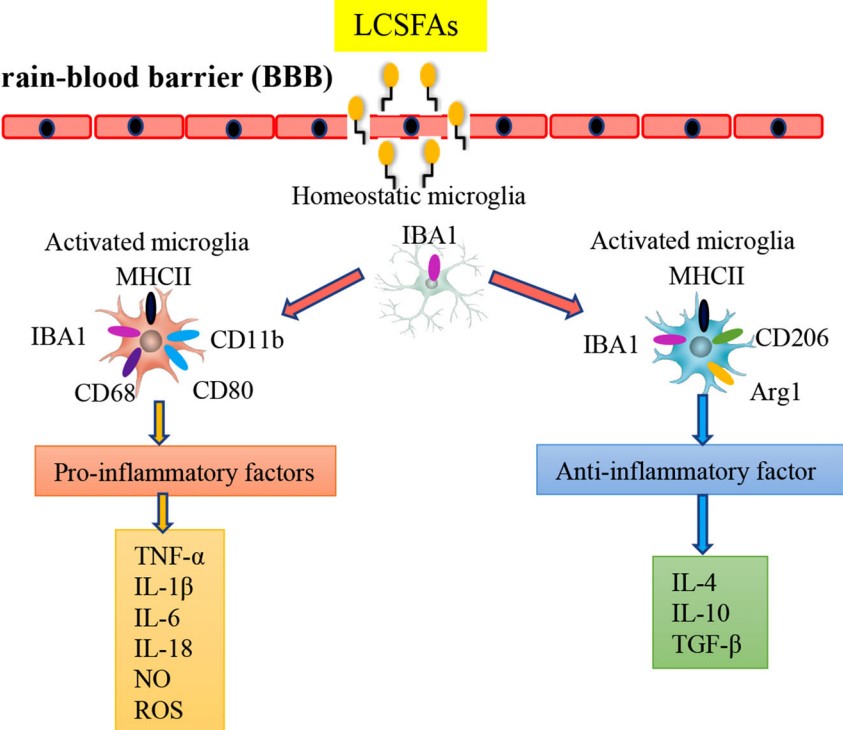

**Figure 2. Long chain saturated fatty acids (LCSFAs) induce activation of microglia contributing to neuroinflammation**

Microglia polarize toward pro-inflammatory or anti-inflammatory phenotypes after exposure to HFD. Activated microglia, characterized by the expression of ionized calcium-binding adaptor molecule 1 (IBA1), major histocompatibility complex II (MHCII), CD68, CD80 and CD11b, release pro-inflammatory factors including interleukin (IL)-1$\beta$, IL-6, IL-18, nitric oxide (NO), tumour necrosis factor-$\alpha$ (TNF-$\alpha$), reactive oxygen species (ROS) and proteases. Activated microglia show high expression of IBA1, MHCII, arginase-1 (Arg1) and CD206, and release anti-inflammatory cytokines including IL-10, IL-4 and transforming growth factor $\beta$ (TGF-$\beta$).

LCSFAs increases brain–blood barrier (BBB) permeability and induces BBB dysfunction (D'Alonzo et al., 2020). With an anatomical proximity to the BBB, the hypothalamus is the front-line brain region that would be initially affected by any detrimental factors that are leaking through a dysfunctional BBB (Thaler et al., 2012). Hypothalamic inflammation often occurs in animals fed a HFD, and the transcription of inflammatory genes is activated as early as a few hours after the first exposure to a HFD (Thaler et al., 2012). In addition, neurogenesis in the hypothalamus of adult mice exposed to prolonged HFD feeding was significantly attenuated, which may be secondary

to HFD-induced neuroinflammatory responses. Evidence supports a multi-factorial explanation of HFD-induced hypothalamic inflammatation, including infiltration of peripheral macrophages (Chen et al., 2021; Lainez et al., 2018), stimulation of resident microglia (Baufeld et al., 2016; Milanski et al., 2009; Zhang et al., 2008) and direct astrocyte activation. Among these, microglia have gained the most attention due to their local residential advantages.

Microglial inflammatory activation in the mediobasal hypothalamus (MBH) following HFD feeding is associated with decreased number of

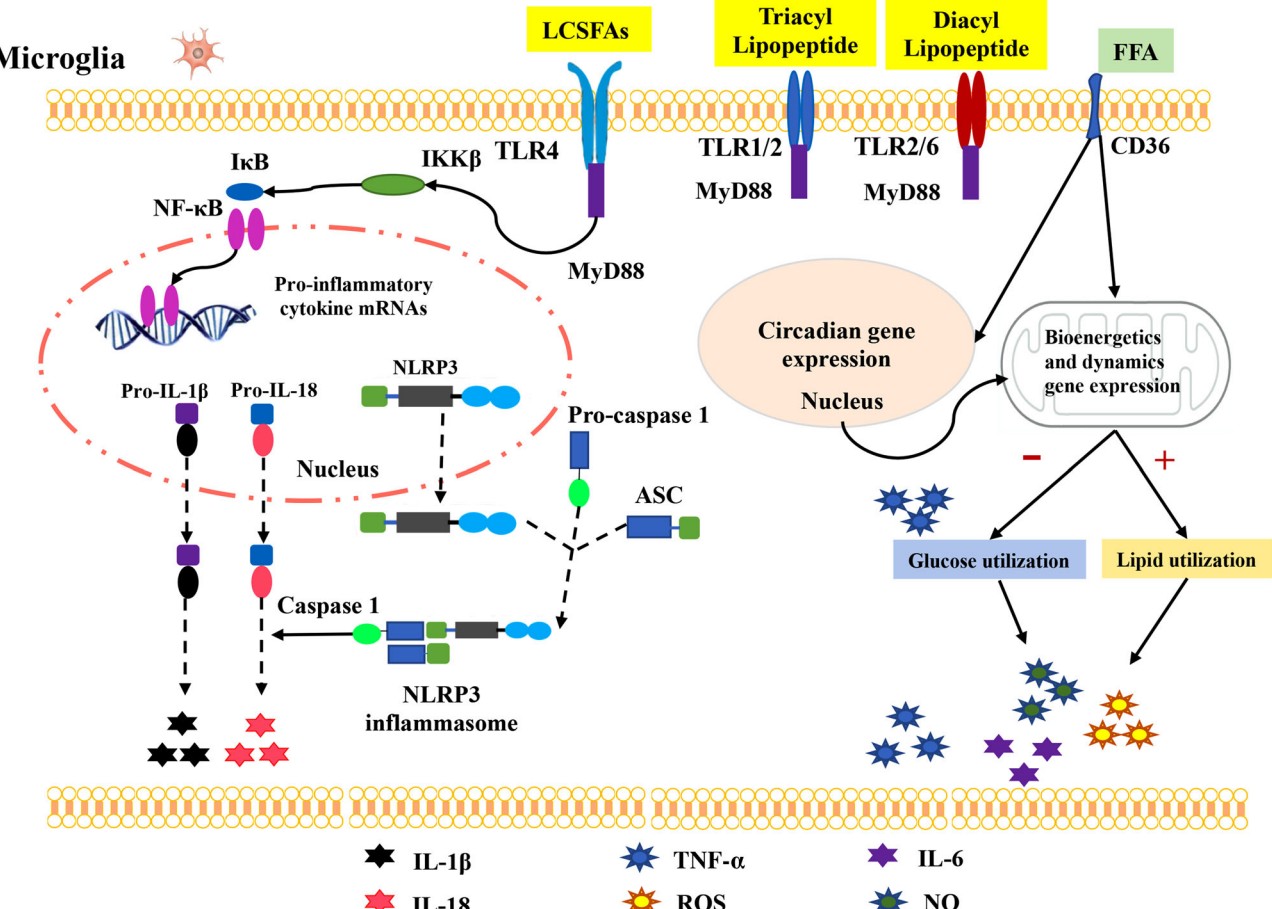

**Figure 3. The three possible signalling pathways of HFD-induced hypothalamic inflammation in microglia activation**
(1) The long chain saturated fatty acids (LCSFAs) in HFD activate microglia via the MyD88-dependent Toll-like receptor 4 (TLR4)/nuclear factor κB (NF-κB)/NLRP3 inflammasome signalling pathway. Activated NF-κB translocates to the nucleus and encodes pro-interleukin (IL)-1β and pro-IL-18. NLRP3 inflammasome interaction with ASC through CARD–CARD results in recruitment of pro-caspase 1, which is then converted to caspase 1. Activated caspase 1 cleaves and activates pro-IL-1β and pro-IL-18 into IL-1β and IL-18, respectively. Triacyl lipopeptides activate TLR1/2 dimmers. TLR2 forms dimmers with TLR6 and mediates responses to diacyl lipopeptides. (2) Free fatty acid (FFA) binds to receptor CD36 into microglia and is transported to the nucleus, where its increase leads to loss of circadian gene expression in microglia, impairs microglia's immunometabolic function and activates microglia. The persistently activated microglia secrete tumour-necrosis factor-α (TNF-α). (3) Meanwhile, the increased FFA and TNF-α enhance the expression of mitochondrial bioenergetics and kinetics genes during the light phase, resulting in decreased glucose utilization and increased lipid utilization, ultimately leading to production of a reactive oxygen species (ROS), TNF-α, IL-6 and nitric oxide (NO).

pro-opiomelanocortin (POMC) neurons and impairment of hypothalamic adult neural stem cells (Li, Tang et al., 2012; McNay et al., 2012; Yi et al., 2017). Researchers have shown that in HFD-induced hypothalamic inflammation, microglia change their appearance, characterized by prolonged processes emanating from a discretely enlarged cell body and molecular signature (Baufeld et al., 2016). Of interest, Yi et al. (2017) observed that microglia perform their function in a circadian pattern driven by nutrient intake and exhibit higher activity in the dark, active phase compared to the light, sleep phase under physiological conditions in mice. However, the hypothalamic microglia of mice fed a HFD lose their circadian rhythm and continued to activate (Yi et al., 2017). Loss of the daily rhythm in microglial circadian genes is accompanied by alterations in substrate utilization and energy production, which may impair microglial immunometabolic function.

Therefore, HFD appears to affect hypothalamic microglial cells at multiple levels, ranging from shaping phenotypic characteristics to altering functional properties. Whether these changes have impacts on other hypothalamic neuronal populations that are responsible for growth (growth hormone-releasing hormone neurons) or reproduction (GnRH neurons) requires further investigation.

**Possible signalling pathways of HFD-induced hypothalamic inflammation in microglia activation.** So far, the molecular mechanisms underlying HFD-induced hypothalamic inflammation by abnormally activated microglia are not fully understood. Toll-like receptors (TLRs) are a class of transmembrane protein receptors that recognize pathogen- or damage-associated molecular patterns, and are involved in the development of various CNS neuroimmune diseases (Kumar, 2019). TLR1 to TLR9 are located on the microglial surface and are key receptors for activation of microglia to release ROS, NO and proinflammatory cytokines (Bsibsi et al., 2002; Olson & Miller, 2004). In addition, studies in brain tissue slice cultures have confirmed that TLR-activated microglia induce neuronal injury through oxidation products and proinflammatory cytokine release (Duport & Garthwaite, 2005; Papageorgiou et al., 2016; Schilling et al., 2021). It is known that TLR1, 2, 4 and 6 can recognize lipid-containing sequences. Triacyl lipopeptides activate TLR1/2 dimers. TLR2 forms dimers with TLR6 and mediates responses to diacyl lipopeptides (Lim & Staudt, 2013) (Fig. 3). TLR4 is activated by lipopolysaccharide (LPS), which is widely used to investigate the molecular signalling mechanism of microglial activation in neuroinflammation (Jack et al., 2005; Olson & Miller, 2004; Papageorgiou et al., 2016).

Previous studies have suggested a role of the TLR4/NF-$\kappa$B signalling pathway in microglial injury (Muhammad et al., 2019). As a receptor for LPS and saturated and polyunsaturated fatty acids, TLR4 has been identified as being expressed on the microglial membrane and mediates many neuroinflammatory diseases (Milanski et al., 2009; Nie et al., 2018). TLR4 loss-of-function mutation or pharmacological inhibition of TLR4 alleviates HFD-induced weight gain and attenuates HFD-induced high expression of hypothalamic inflammation factors such as IL-6, IL-1$\beta$ and TNF-$\alpha$ (Milanski et al., 2009). As a transcriptional factor complex, NF-$\kappa$B is located downstream of the TLR4 signalling pathway and is also a first-line molecules that initiates the innate immune response (Wang et al., 2012). The inhibition of the TLR4/NF-$\kappa$B signalling pathway in microglia by means of pharmacological or antibody-based blocking agents improves HFD-induced neuroinflammation (Jiao et al., 2018; Lee et al., 2012; Muhammad et al., 2019; Zhang et al., 2020; Zusso et al., 2019). Previous researches have revealed that alleviating neuronal inflammation by inhibiting the TLR4/NF-$\kappa$B pathway restores hypothalamic control of energy homeostasis, thereby reducing susceptibility to HFD-induced obesity (Benzler et al., 2015; Kleinridders et al., 2009; Valdearcos et al., 2018; Zhang et al., 2008). Thus, it is likely that TLR4/NF-$\kappa$B signalling may exert an important role in HFD-induced microglial activation, contributing to neuronal inflammation (Fig. 3).

**HFD induces a shift in substrate utilization and microglial function.** Consuming a HFD disturbs microglial daily rhythmicity and stimulates microglial reactivity persistently in the MBH. The persistently activated microglia secrete TNF-$\alpha$ (Yi et al., 2017). TNF-$\alpha$ stimulates mitochondrial ATP production promoting mitochondrial stress in mouse primary hypothalamic neurons. Moreover, TNF-$\alpha$ induces mitochondrial elongation in neurites of MBH neurons and drives neuronal energy requirements by increasing neuronal firing rates (Yi et al., 2017). In the long run, the persistently induced mitochondrial stress could contribute to key neuronal dysfunction. It is well-known that the daily rhythms generated in behavioural, physiological and hormonal processes in mammals enable animals to adapt to daily environmental changes and optimize metabolic function throughout the day. The findings of Yi et al. show that HFD causes the loss of circadian rhythm genes in microglia and impairs microglial immunometabolic function, mainly during the transition period between dark and light, accompanied by substrate utilization and energy production changes (Milanova et al., 2019). Changes in substrate utilization are known to affect the activation status of microglia. Sudden changes in energy utilization promote a homeostatic state or an activated state (Lee et al., 2018; O'Neill et al., 2016) (Fig. 3).

Substrate utilization of microglia is based on glutamate, glucose and fatty acid metabolism (Bernier, York, MacVicar et al., 2020; O'Neill et al., 2016). Yi and colleagues have shown that HFD feeding reduces glutamate and glucose utilization by microglia during the active period of the dark phase. However, a HFD causes microglia to increase lipid utilization and perception during the light phase, suggesting a switch to fatty acid utilization during sleep in rats. Apart from changes in substance utilization, a HFD also increases the expression of mitochondrial bioenergetics and kinetics gene during the light phase in rats (Milanova et al., 2019) (Fig. 3). These changes seem to be specifically assigned to microglial immunometabolism given that assessment of functional gene expression in monocytes shows little or no effect of a HFD on blood monocyte immunometabolism (Milanova et al., 2019). In addition, a HFD has been shown to increase the expression of the key mitochondrial protein uncoupling protein 2 in hypothalamic microglia, which induces microglial activation and inflammation in the arcuate nucleus (ARC) (Kim et al., 2019). Therefore, a HFD may alter microglial metabolism and function by shifting substrate utilization and modifying mitochondrial gene expression to promote a proinflammatory state, causing proximal neuronal dysfunction.

**Hypothalamic inflammation may be an underlying pathophysiological mechanism of HFD-induced ovulation disorders.** Hypothalamic GnRH neurons are key neurons that regulate the reproductive system by generating pulsatile GnRH secretion. The majority of GnRH neurons are located in ARC (part of the MBH and in the medial preoptic area), and their pulse-generating function is closely regulated by several other neurons in the hypothalamus. Kisspeptin neurons are mainly located in ARC and the anteroventral periventricular nuclei. Given the proximity, kisspeptin is known to activate GnRH neurons to stimulate GnRH secretion (Rumpler et al., 2020). Neurons expressing kisspeptin, neurokinin B (NKB) and dynorphin A in ARC are called KNDy (kisspeptin/NKB/Dyn) neurons (Bartzen-Sprauer et al., 2014; Goodman et al., 2013; Navarro et al., 2009). The KNDy neurons generate discrete neural signals and act in ARC to control the activity of the GnRH/LH pulse generator. NKB is the positive regulator of this circuitry, and Dyn is responsible for GnRH pulse termination (Ikegami et al., 2022; Mittelman-Smith et al., 2016; Nagae et al., 2021; Wakabayashi et al., 2010). The metabolic state of KNDy neurons determines neuropeptide expression (Ziarniak et al., 2020). The activity of KNDy neurons is influenced by the balance of negative energy (undernutrition) and positive energy (overnutrition). Moreover, evidence demonstrates

that KNDy neurons in ARC are locally connected to agouti-related protein (AgRP)/neuropeptide Y (NPY) and POMC/CART neurons (Backholer et al., 2010; True et al., 2013). Neuropeptides secreted by AgRP/NPY and POMC/CART neurons affect GnRH pulse release (Fig. 4).

Given the plastic property of KNDy neurons in response to metabolic challenges, it is of interest to know the influence of a HFD on KNDy neuronal peptide expression in ARC. Yang et al. revealed that HFD feeding had no effect on arcuate *Tac2* mRNA (which encodes NKB) expression in female mice. However, they reported that the expression of *Kiss1*, *Kiss1r* and NKB receptor mRNA (*Tacr3*) was reduced in ARC of HFD-fed female mice (Yang et al., 2016). The data of Ziarniak et al. (2018) demonstrated that HFD had no effects on the number of Kiss1, Tacr3 and Dyn neurons in ARC of female mice. Studies by other groups of Dyn expression in ARC in HFD-fed female rodents corroborate this result (Minabe et al., 2021; Yang et al., 2016; Ziarniak et al., 2020).

Although evidence regarding the effect of HFD on neuronal peptide expression is inconclusive, it is generally accepted that KNDy neurons mediate oestrogen negative feedback on LH secretion, relay progesterone inhibition of pulsatile GnRH secretion and modulate the pulsatile release of GnRH and that HFD can affect KNDy neuron activity and disrupt the oestrous cycle (Li, Lin et al., 2012). Although it is still unclear what exact mechanism mediates the interaction between a HFD and KNDy neurons, indirect evidence supports the inflammatory theory to some extent. It has been noticed that hypothalamic inflammation occurs well before the development of weight gain and obesity following HFD consumption. This is thought to be independent of the obese phenotype as similar results were seen in the lean group. Activated microglia play a vital role in the progression of LPS- or HFD-induced hypothalamic inflammation (Kim et al., 2019; Valdearcos et al., 2014; Zusso et al., 2019). Coincidently, Fergani et al. (2017) found that administration of LPS decreased the percentage of activated kispepetin and Dyn immunoreactive cells and then inhibited the onset of the LH surge. Moreover, previous studies have shown that hypothalamic neurogenesis occurs in adult mice and that HFD consumption inhibits generation of new neurons in the hypothalamic ARC (Bless et al., 2016; Li, Tang et al., 2012; McNay et al., 2012). This hints at a link of the form HFD–activation of microglia–hypothalamic inflammation–dysfunction of KDyn neurons–inhibition of GnRH release (Fig. 4).

Clinically, many studies have identified that a high carbohydrate diet and high saturated fat ingestion are associated with inflammation, insulin resistance, androgen excess and high incidence of anovulation in women with PCOS (Gonzalez et al., 2014, 2020). By contrast, anti-inflammatory dietary intervention

contributes to improving insulin sensitivity, promoting a physiological menstrual cycle and improving fertilization in PCOS women (Barrea et al., 2019; Salama et al., 2015). Thus, hypothalamic inflammation is likely to be an underlying mechanism for the development of PCOS, and drugs that can improve inflammation may be a potential therapeutic option.

## SGLT2 inhibitors improve female reproductive function

SGLT2 inhibitors lower plasma glucose by inhibiting renal tubular absorption of glucose. There are several SGLT2 inhibitors used in diabetes treatment (empagliflozin, dapagliflozin, canagliflozin, etc.). Empagliflozin and dapagliflozin are highly selective SGLT2 inhibitors that have been shown to give not only glycaemic control but also weight loss and improved insulin sensitivity. The direct effect of SGLT2 inhibitors on female reproductive functions has not been fully studied. In clinical practice, there are only a small number of studies that have evaluated the effect of SGLT2 inhibitors in PCOS patients without diabetes. A 12-week, randomized open-label controlled trial was performed in obese women with PCOS. Participants were randomized to either empagliflozin at a dose of 25 mg daily or metformin 1500 mg. The result demonstrated significant decreases in weight, hip circumference, body mass index, waist circumference, basal metabolic rate and fat mass in the empagliflozin group compared with the

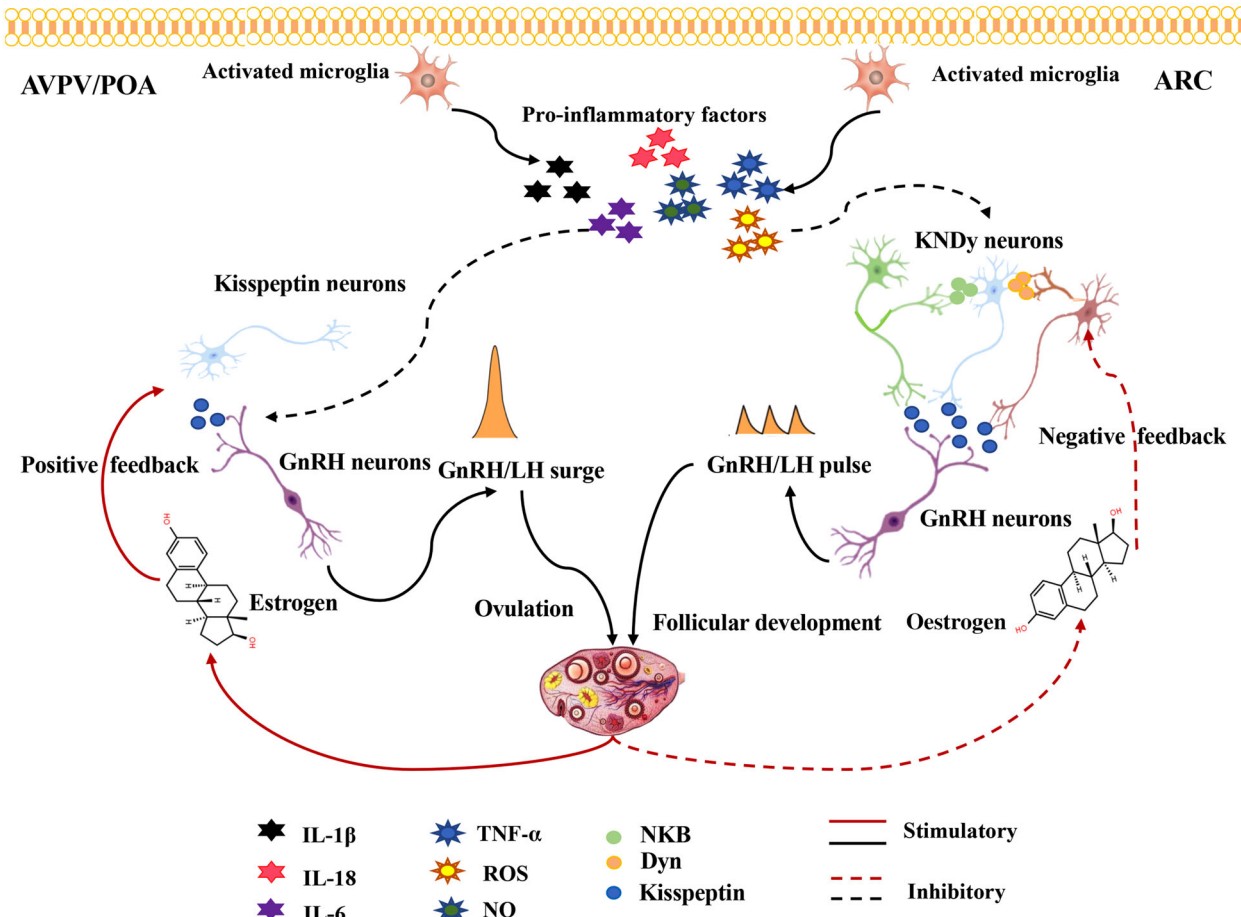

**Figure 4. Hypothalamic inflammation may be an underlying pathophysiological mechanism of HFD-induced HPO axis dysfunction**
In ARC, the kisspeptin neurons express both neurokinin B (NKB) and dynorphin A and are called KNDy (kisspeptin/NKB/Dyn) neurons. NKB stimulates kisspeptin release through KNDy neurons and Dyn inhibits kisspeptin release via KNDy neurons. The role of KNDy neurons is that of controlling gonadotropin-releasing hormone (GnRH)/luteinizing hormone (LH) pulse release, which is responsible for follicular development and oestrogen negative feedback. The kisspeptin neurons located in the anteroventral periventricular nuclei (AVPV) controlling GnRH/LH surge release are responsible for ovulation and oestrogen positive feedback. Pro-inflammatory factors, such as TNF-α, IL-1β, IL-18, IL-6, NO and ROS, impair neurogenesis and increase neurodegeneration.

metformin group. But there were no significant changes in fasting glucose, insulin, androgen, and hs-CRP after empagliflozin treatment (Javed et al., 2019). Another study in obese women with PCOS has demonstrated that dapagliflozin 10 mg daily for 24 weeks resulted in significant improvements in blood glucose, insulin sensitivity and secretion, and reductions in weight, abdominal adiposity, waist circumference, body mass index, testosterone and free androgen (Elkind-Hirsch et al., 2021).

Both SGLT1 and SGLT2 are expressed in the renal proximal tubule and are together responsible for glucose reabsorption (accounting for ~3% and ~97%, respectively). SGLT1 is also expressed in the small intestine and plays an important role in intestinal glucose uptake and secretion of glucagon-like peptide-1. Therefore, dual inhibitors of SGLT1 and SGLT2 are appealing drugs for treatment of diabetes and obesity. Licogliflozin is a dual inhibitor of SGLT1/2 and is currently in clinical trials. A recent randomized, double-blind, 2-week trial in PCOS patients showed that licogliflozin reduced hyperinsulinaemia by 70%, androstendione by 19% and dehydroepiandrosterone sulphate by 24% (Tan et al., 2021). However, none of these studies measured the effects of SGLT2 inhibitors on the menstrual cycle, ovulatory disorder or fertility in PCOS women.

In a dihydrotestosterone-induced PCOS rat model, empagliflozin for 3 weeks decreased fat mass, plasma leptin and blood pressure, but there was no improvement of hyperinsulinaemia and no impact on HbA1c and lipid profile (Pruett et al., 2021). In our animal study, we found that female melanocortin-4 receptor-deficient obese mice (MC4R KO) had a metabolic disorder similar to PCOS patients and demonstrated irregular oestrous cycles and ovulation failure (Chen et al., 2017). We also showed that 14-week-old MC4R KO mice treated with dapagliflozin for 11 weeks had significantly reduced body weight gain, normalized oestrous cycle, recovered pulsatile LH secretion and restored ovulation (Cui et al., 2022). Meanwhile, expression of *Kiss1* mRNA and *Gnrh* mRNA in hypothalamus of MC4R KO mice was increased after dapagliflozin treatment. Thus, we provide evidence that SGLT2 inhibitor is able to improve ovulatory dysfunction, at least, in MC4R KO mice and this may be mediated by revival of the hypothalamic regulating neurons that are responsible for the HPO axis.

### The possible mechanism of SGLT2 inhibitors that ameliorate HPO axis dysfunction

To explore the mechanism of SGLT2 inhibitors on the HPO axis, it is reasonable to study SGLT2 expression. Though SGLT2 is not found in ovaries, increasing evidence has reported SGLT2 expression in CNS regions such as the hippocampus, hypothalamus and BBB endothelial cells (Chiba et al., 2020). SGLT2 inhibitors are lipid-soluble drugs and can traverse the BBB and reach the CNS. This is supported by evidence that SGLT2 inhibitors have neuroprotective properties in patients and animals with T2D, such as improving impaired cognitive function (Hayden et al., 2019; Hierro-Bujalance et al., 2020; Lin et al., 2014), and have further applications in many neurodegenerative diseases such as AD, PD, Huntington's disease, epilepsy and cerebral ischaemia (Arab et al., 2021; El-Sahar et al., 2020; Hierro-Bujalance et al., 2020; Lin et al., 2014). Recently, a clinical study from Hong Kong reported that the use of SGLT2 inhibitor is associated with a decreased risk of PD (Mui et al., 2021).

The neuroprotective effects of SGLT2 inhibitors are likely to be by attenuating oxidative stress, reducing inflammation and improving immune response in which microglia may play an important role. The bulk of the evidence suggests that microglia-mediated neuro-inflammation contributes to pathology and progression of AD (Santiago et al., 2021; Subbarayan et al., 2021). It has been shown that microglial activation induced by systemic inflammation via the NLRP3 inflammasome impairs microglial clearance, contributing to development and progression of AD (Tejera et al., 2019). The latest clinical research has demonstrated that treatment with empagliflozin inhibits NLRP3 inflammasome activity and IL-1$\beta$ secretion in diabetic patients at high risk of cardiovascular disease (Kim et al., 2020). HFD-induced inflammation in hypothalamus, as indicated by increased numbers of IBA1-positive microglia and IL-6 mRNA levels, is decreased in canagliflozin-treated mice (Naznin et al., 2017). Dapagliflozin also exerts anti-inflammatory effects in the brain by preventing the phosphorylation of NF-$\kappa$B, and restores cognitive function in HFD-induced obese rats (Sa-Nguanmoo et al., 2017). Microglia are brain-resident innate immune cells whose function and phenotype are shaped by their cellular metabolism. The nutrient demands of microglia depend on changes in their microenvironment. Recent studies have confirmed that most of the energy supply of microglia is dependent on glucose and its oxidative phosphorylation. Disturbances in the body's nutrient availability can affect the energy levels of microglia, thereby affecting their phenotypic states and immune function. Several studies have shown that microglia can also metabolize amino acids, pyruvate, lactate, ketone bodies and fatty acids (Bernier, York, Kamyabi et al., 2020; Kalsbeek et al., 2016; Monsorno et al., 2022). When glucose is in short supply, microglia can metabolize glutamine to promote the tricarboxylic acid cycle and maintain mitochondrial respiration (Bernier, York, Kamyabi et al., 2020). Milanova et al. (2019) were the first to report that a HFD decreased microglial glucose utilization and oxidation, resulting in increased lipid sub-

strate utilization. In the brain, glucose utilized by cells is transported by glucose transporters (i.e. GLUTs) and SGLTs. Lines of evidence demonstrate that glucose is transported into brain cells by active transport via SGLT pathways and accumulates in those cells. *In vitro*, using $\alpha$-methyl-4-[$^{18}$F]fluoro-4-deoxy-D-glucopyranoside autoradiography to probe SGLT expression responsible for glucose transport in the brain, we have confirmed that SGLT2 is localized in microglial cell bodies with immunofluorescence (Yu et al., 2013). Therefore, we predict that inhibition of SGLT2 should change brain energy homeostasis and disturb microglial metabolic substrates and that metabolic reprogramming modifies the microglial phenotype. We also expect that SGLT2 inhibitors impacts microglial metabolism and immune function, preventing HFD-induced hypothalamic inflammation, impaired neurogenesis and neurodegeneration.

In the *db/db* T2D mouse, B. Lin and colleagues report that empagliflozin treatment increases brain-derived neurotrophic factor, the decline of which is associated

with cognitive dysfunction (Franzmeier et al., 2021), and decreases cerebral oxidative stress, which is association with cognitive impairment as well (Lin et al., 2014). In a PD rat model, dapagliflozin can reduce oxidative stress in neurons by decreasing lipid peroxides, and consequently restoring the impaired DJ/Nrf2 pathway. Furthermore, dapagliflozin prevents ROS-dependent neuronal apoptosis via the phosphoinositide 3-kinase/AKT/glycogen synthase kinase 3 $\beta$ (Ser9) pathway. In addition, dapagliflozin inhibits neuroinflammation by suppressing NF-$\kappa$B signalling pathway activation (Arab et al., 2021). A small number of studies have shown that cerebral atherosclerosis is also an important contributor to the increased prevalence of AD independent of ischaemia (Bos et al., 2015; Wingo et al., 2020). Experimental evidence suggests that SGLT2 inhibitors alleviate atherosclerosis by reducing inflammation, improving endothelial function, inhibiting the activity of the renin–angiotensin–aldosterone system, and modulating

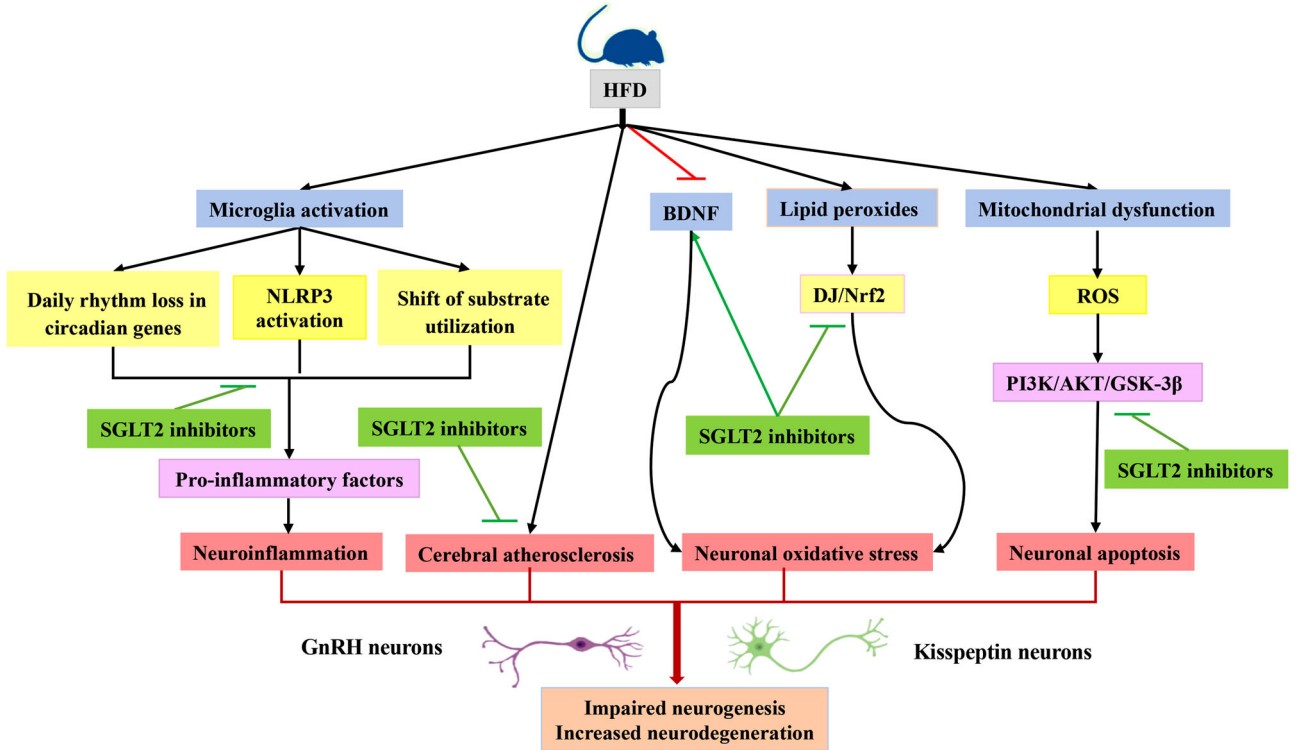

**Figure 5. The possible mechanism by which SGLT2 inhibitors ameliorate HPO axis dysfunction induced by HFD**

According to the evidence, SGLT2 inhibitors have neuroprotective properties via the following four pathways. (1) Inhibition of microglial activation: suppression of the phosphorylation of NF-$\kappa$B; controlling NLRP3 inflammasome activity; and shifting substrate utilization, resulting in decreasing proinflammatory factor release and neuroinflammation. (2) Alleviation of cerebral atherosclerosis. (3) Increasing brain-derived neurotrophic factor (BDNF), decreasing lipid peroxides through restoring impaired DJ/Nrf2 pathway, which reduce oxidative stress of neuron. (4) Curbing the phosphoinositide 3-kinase (PI3K)/AKT/glycogen synthase kinase 3 $\beta$ (GSK-3$\beta$) (Ser9) pathway, inhibiting ROS-dependent neuronal apoptosis. Together, all these events ultimately ameliorate impaired neurogenesis and increased neurodegeneration.

macrophage polarization (Day et al., 2020; Ganbaatar et al., 2020; Koyani et al., 2020; Liu et al., 2021; Nakatsu et al., 2017; Park et al., 2020).

These findings suggest that SGLT2 inhibitor prevention of cognitive decline in those models is related to the reduction in cerebral oxidative stress, alleviation of cerebral atherosclerosis and suppression of microglia-mediated neuroinflammation. As many HFD-induced pathological changes resemble neurodegenerative diseases, we speculate that the underlying mechanism for the beneficial effects of the SGLT2 inhibitors on HFD-induced HPO axis disorders and/or PCOS may be similar to that of improving cognitive decline (Fig. 5). SGLT2 inhibitors may rescue the impaired GnRH surge in females and restore ovulatory function, as well as improve body weight and insulin resistance induced by a HFD.

## Conclusions

HFD intake has a detrimental effect on HPO axis function and female fertility that is independent of the obese phenotype. Previous studies have shown that a HFD activated microglia via the TLR4/NF-$\kappa$B signalling pathway, leading to hypothalamic inflammation and impairment of KNDy neuron activity, consequently disrupting the oestrous cycle. However, the molecular mechanism of HFD-induced female reproductive dysfunction remains largely unknown due to limited relevant research. Inspired by mechanistic studies in AD and PD patients by which SGLT2 inhibitors have a positive impact in cognitive decline and neuronal apoptosis, we hypothesize that the possible mechanisms underlying the beneficial effect of SGLT2 inhibitors on the HPO axis may be attributed to reduction in brain oxidative stress and amelioration of hypothalamic neuroinflammation. Based on the fact that a HFD induces HPO axis dysfunction, this review has addressed two questions: (1) Does hypothalamic inflammation contributes to the aetiology of infertility at hypothalamic levels? (2) Is it possible to consider SGLT2 inhibitors as a potent therapy for PCOS and what may be the relevant mechanisms? Although the full landscape of the mechanism underlying HFD-induced hypothalamic HPO axis dysfunction is limited due to lack of adequate research studies, this review is intended to initiate further discussion and attract more valuable research studies in the future.

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

## Additional information

### Competing interests

None.

### Author contributions

Conception or design of the work: X.C., L.H., C.C. Acquisition or analysis or interpretation of data for the work: X.C., L.C., Z. X., C.C. Drafting the work or revising it critically for important intellectual content: X.C., L.H., X.X., C.C. All authors have read and approved the final version of this manuscript and agree to be accountable for all aspects of the work in ensuring that questions related to the accuracy or integrity of any part of the work are appropriately investigated and resolved. All persons designated as authors qualify for authorship, and all those who qualify for authorship are listed.

### Funding

None.

### Acknowledgements

Open access publishing facilitated by The University of Queensland, as part of the Wiley – The University of

Queensland agreement via the Council of Australian University Librarians.

## Keywords

high-fat diet, HPO axis, hypothalamic inflammation, microglia, ovulatory disorder, SGLT2 inhibitor

## Supporting information

Additional supporting information can be found online in the Supporting Information section at the end of the HTML view of the article. Supporting information files available:

**Peer Review History**

