## [Peer Review History · The Journal of Physiology]

SGLT2 inhibitor ameliorates high-fat diet-induced hypothalamic-pituitary-ovarian axis disorders

Xiaolin Chen, Lili Huang, Ling Cui, Zhuoni Xiao, Xiaoxing Xiong, and Chen Chen
DOI: 10.1113/JP283259

Corresponding author(s): Chen Chen (chen.chen@uq.edu.au)

The following individual(s) involved in review of this submission have agreed to reveal their identity: GN Bader (Referee #2)

Review Timeline:

Submission Date:	06-May-2022
Editorial Decision:	14-Jun-2022
Revision Received:	25-Jul-2022
Editorial Decision:	09-Aug-2022
Revision Received:	14-Aug-2022
Accepted:	16-Aug-2022

Senior Editor: Laura Bennet

Reviewing Editor: Rebecca Simmons

Transaction Report:

Dear Professor Chen,

Re: JP-TR-2022-283259 "SGLT2 inhibitor ameliorates high-fat diet-induced hypothalamic-pituitary-ovarian axis disorders" by Xiaolin Chen, Lili Huang, Ling Cui, Zhuoni Xiao, Xiaoxing Xiong, and Chen Chen

Thank you for submitting your Topical Review to The Journal of Physiology. It has been assessed by a Reviewing Editor and by 2 expert referees and I am pleased to tell you that it is considered to be acceptable for publication following satisfactory revision.

The reports are copied at the end of this email. Please address all of the points and incorporate all requested revisions, or explain in your Response to Referees why a change has not been made.

NEW POLICY: In order to improve the transparency of its peer review process The Journal of Physiology publishes online as supporting information the peer review history of all articles accepted for publication. Readers will have access to decision letters, including all Editors' comments and referee reports, for each version of the manuscript and any author responses to peer review comments. Referees can decide whether or not they wish to be named on the peer review history document.

I hope you will find the comments helpful and have no difficulty in revising your manuscript within 4 weeks.

Your revised manuscript should be submitted online using the links in Author Tasks Link Not Available. This link is to the Corresponding Author's own account, if this will cause any problems when submitting the revised version please contact us.

You should upload:

- A Word file of the complete text (including any Tables);
- An Abstract Figure, (with accompanying Legend in the article file)
- Each figure as a separate, high quality, file;
- A full Response to Referees;
- A copy of the manuscript with the changes highlighted.
- Author profile. A short biography (no more than 100 words for one author or 150 words in total for two authors) and a portrait photograph of the two leading authors on the paper. These should be uploaded, clearly labelled, with the manuscript submission. Any standard image format for the photograph is acceptable, but the resolution should be at least 300 dpi and preferably more.

- A 'Cover Art' file for consideration as the Issue's cover image;
- Appropriate Supporting Information (Video, audio or data set https://jp.msubmit.net/cgi-bin/main.plex?form_type=display_requirements#supp).

To create your 'Response to Referees' copy all the reports, including any comments from the Senior and Reviewing Editors into a Word, or similar, file and respond to each point in colour or CAPITALS. Upload this when you submit your revision.

I look forward to receiving your revised submission.

Yours sincerely,

Professor Laura Bennet
Senior Editor
The Journal of Physiology
<https://jp.msubmit.net>
<http://jp.physoc.org>
The Physiological Society
Hodgkin Huxley House
30 Farringdon Lane
London, EC1R 3AW
UK
<http://www.physoc.org>
<http://journals.physoc.org>

EDITOR COMMENTS

Reviewing Editor:

I think the Authors can satisfactorily address the reviewers' concerns, but both reviewers asked for major revisions, all of which need to be addressed. In particular, reviewer 2 raised several important points.

Please also see 'Required Items' below.

REFeree COMMENTS

Referee #1:

Chen et al. (JP-TR-2022-283259) provide a review about the effects of SGLT2 inhibitors on high-fat diet-induced hypothalamic-pituitary-ovarian axis disorders. The topic is timely and of general interest for physiologists, pathophysiologists and researchers in other disciplines. The manuscript partially lacks clarity and scientific depth, however. The authors might consider the following criticism to improve clarity and to strengthen the conclusions.

Major points:

1. Page 8 (PDF). Sentence, 'Thus, HFD causes HPO axis dysfunction traversing multiple levels, from the central regulating neurons down to the peripheral ovarian glands.' Dysfunction in 'central regulating neurons' is not that obvious in the main text. Please, provide more details and/or rephrase.
2. Page 10, Figure 2 (legend) and elsewhere. Sentence, 'The inflammatory cytokines include tumor necrosis factor- α (TNF- α), interleukin-1 β (IL-1 β), interleukin-18 (IL-18), superoxide, nitric oxide (NO), reactive oxygen species (ROS), and proteases.' ROS and RNS are not cytokines. What is the difference between 'superoxide' and 'ROS'? Please, check and rephrase.
3. Page 11, Figure 2 and elsewhere. The terminology of 'resting' microglia is obsolete. Please, use 'surveying' or 'homeostatic' microglia.
4. Page 11 and Figure 2. Please, differentiate between general markers and activation markers of microglia.
5. Page 11. The observation (concept) of 'over-reactive microglia' needs to be discussed in more detail (experimental settings and models, stimuli/triggers etc.).
6. Page 14 and Figure 3. Sentence, 'Previous studies suggest that a Toll-like receptor 4 (TLR4)/ nuclear factor kappa-light-chain-enhancer of activated B cells (NF- κ B) signaling pathway in microglia injury (Muhammad et al., 2019).'

 - a) Is 'microglia-mediated injury' meant? Please, check and rephrase.
 - b) Activation of TLR-4 in microglia is indeed proinflammatory. However, many studies have shown that sole TLR-4 activation in microglia is not sufficient to induce neuronal death (e.g., Chao et al., J Immunol, 1992; Duport and Garthwaite, Neuroscience, 2005; Papageorgiou et al., Proc Natl Acad Sci U S A, 2016; Schilling et al., Brain Behav Immun, 2021). Please, discuss both context and effects of TLR-4 activation in microglia in more detail.

7. Pages 19 ff and Figure 4.
 - a) Please, discuss the role of microglia in 'hypothalamic inflammation' in more detail.
 - b) Please, include microglia and its inflammatory release products in Figure 4.

8. Page 21 and elsewhere. Please, introduce the features of the drug empagliflozin. This also applies to other drugs discussed.

9. Pages 22 ff and Figure 5. It is not really clear how SGLT2 inhibitors might attenuate activation of microglia that might proliferate and become glycolytic (e.g., Gosh et al., *Glia*, 2018; Bernier et al., *Trends Neurosci*, 2020). Please, discuss this issue in more detail.

10. Figure 5. Please, make clear which pathways relate to microglia and which not.

Minor points:

11. Abbreviations. Please, generally reduce the use of abbreviations for the sake of readability (e.g., GS, ARC, NKB, CA etc.). Please, also reduce the use of uncommon and less intuitive abbreviations (e.g., SGLT2i, VO etc.).

12. Page 8. What are 'master neurons'? Please, rephrase.

13. Page 8. Sentence, '(...) high levels of pro-inflammatory cytokines and increased macrophage infiltration (...)'. Please, specify location (tissue and/or layer).

14. Page 13. Sentence, 'Microglial inflammatory activation in the medio-basal hypothalamus (MBH) following HFD-feeding is associated with (...)'. Please, specify 'neuronal functional changes'.

15. Page 13. Sentence, 'Of interest, Tschop et al. have observed that microglial cells exert their function in a strict circadian pattern, (...)'. This is somewhat misleading and likely not true for acute infections etc. Please, rephrase.

16. Figure 4 and legend. How relevant is 'neurogenesis' in the adult hypothalamus? Please, comment on.

Referee #2:

The manuscript "SGLT2 inhibitor ameliorates high-fat diet-induced hypothalamic-pituitary-ovarian axis disorders" envisages role of SGLT2 inhibitors in various disorders (cognitive, PCOD etc) and their "hypothetical" mechanism of action in these conditions, which is attributed to the reduction in brain oxidative stress and the amelioration of hypothalamic neuroinflammation. The authors have build their review on following major points:

1. Consumption of HFD causes hypothalamic-pituitary-ovarian (HPO) axis dysfunction
2. HFD-induced hypothalamic inflammation is involved in HPO axis dysfunction

3. SGLT2i improve female reproductive function
4. The possible mechanism of SGLT2 inhibitor ameliorates HPO axis dysfunction

Overall the manuscript has been written meticulously, with up to date review of literature. Hypothesis put forward has some basis, which can be deliberated upon. However, the manuscript needs proper language editing, as many statements/ points lead to confusion because of language problems. I have highlighted some points in the manuscript that need to be addressed (file attached for reference). Some of the important points that need to be addressed are:

1. "Both obese mice and non-obese mice were subfertility and show impaired ovarian function regardless of the obese phenotype"
- 2."Upon activation, NLRP3 interaction with ASC through CARD-CARD result in recruiting pro-caspase 1, which then pro-caspase 1 is converted into caspase 1".
- 3."TLR4 loss-of-function mutation or pharmacological inhibition of TLR4 alleviates HFD-induced weight gain and attenuates HFD induced hypothalamic highly expression of inflammation factors such as IL-6 and IL-10"
- 4."Yang et al. revealed HFD-fed was no effects in the arcuate Tac2 mRNA (which encodes NKB) expression in female mice".
5. " This somehow, provide hints to form an HFD-hypothalamic inflammation"
- 6."significant decreased body weight gain"
- 7." Black solid arrow, stimulate; red blunt line, inhibit; green solid arrow, stimulating effect of SGLT2i; green blunt line, inhibitory effect of SGLT2i".

REQUIRED ITEMS:

-Please include an Abstract Figure. The Abstract Figure is a piece of artwork designed to give readers an immediate understanding of the Review Article and should summarise the main conclusions. If possible, the image should be easily 'readable' from left to right or top to bottom. It should show the physiological relevance of the Review so readers can assess the importance and content of the article. Abstract Figures should not merely recapitulate other figures in the Review. Please try to keep the diagram as simple as possible and without superfluous information that may distract from the main conclusion of the Review. Abstract Figures must be provided by authors no later than the revised manuscript stage and should be uploaded as a separate file during online submission labelled as File Type 'Abstract Figure'. Please ensure that you include the figure legend in the main article file. All Abstract Figures will be sent to a professional illustrator for redrawing and you may be asked to approve the redrawn figure before your paper is accepted.

-Your MS must include a complete "Additional information section" with the following 4 headings and content:

Competing Interests: A statement regarding competing interests. If there are no competing interests, a statement to this effect must be included. All authors should disclose any conflict of interest in accordance with journal policy.

Author contributions: Each author should take responsibility for a particular section of the study and have contributed to writing the paper. Acquisition of funding, administrative support or the collection of data alone does not justify authorship; these contributions to the study should be listed in the Acknowledgements. Additional information such as 'X and Y have contributed equally to this work' may be added as a footnote on the title page.

It must be stated that all authors approved the final version of the manuscript and that all persons designated as authors qualify for authorship, and all those who qualify for authorship are listed.

Funding: Authors must indicate all sources of funding, including grant numbers. If authors have not received funding, this must be stated.

It is the responsibility of authors funded by RCUK to adhere to their policy regarding funding sources and underlying research material. The policy requires funding information to be included within the acknowledgement section of a paper. Guidance on how to acknowledge funding information is provided by the Research Information Network. The policy also requires all research papers, if applicable, to include a statement on how any underlying research materials, such as data,

samples or models, can be accessed. However, the policy does not require that the data must be made open. If there are considered to be good or compelling reasons to protect access to the data, for example commercial confidentiality or legitimate sensitivities around data derived from potentially identifiable human participants, these should be included in the statement.

Acknowledgements: Acknowledgements should be the minimum consistent with courtesy. The wording of acknowledgements of scientific assistance or advice must have been seen and approved by the persons concerned. This section should not include details of funding.

-Please upload separate high quality figure files via the submission form.

-Author profile(s) must be uploaded via the submission form. Authors should submit a short biography (no more than 100 words for one author or 150 words in total for two authors) and a portrait photograph of the two leading authors on the paper. These should be uploaded, clearly labelled, with the manuscript submission. Any standard image format for the photograph is acceptable, but the resolution should be at least 300 dpi and preferably more. A group photograph of all authors is also acceptable, providing the biography for the whole group does not exceed 150 words.

END OF COMMENTS

Confidential Review

06-May-2022

The Journal of Physiology

<https://jp.msubmit.net>

JP-TR-2022-283259

Title: SGLT2 inhibitor ameliorates high-fat diet-induced hypothalamic-pituitary-ovarian axis disorders

Authors: Xiaolin Chen
Lili Huang
Ling Cui
Zhuoni Xiao
Xiaoxing Xiong
Chen Chen

Author Conflict: No competing interests declared

Author Contribution: Xiaolin Chen: Conception or design of the work; Acquisition or analysis or interpretation of data for the work; Drafting the work or revising it critically for important intellectual content; Final approval of the version to be published; Agreement to be accountable for all aspects of the work Lili Huang: Acquisition or analysis or interpretation of data for the work; Drafting the work or revising it critically for important intellectual content; Final approval of the version to be published; Agreement to be accountable for all aspects of the work Ling Cui: Acquisition or analysis or interpretation of data for the work; Drafting the work or revising it critically for important intellectual content; Final approval of the version to be published; Agreement to be accountable for

Disclaimer: This is a confidential document.

all aspects of the work Zhuoni Xiao: Conception or design of the work; Drafting the work or revising it critically for important intellectual content; Final approval of the version to be published; Agreement to be accountable for all aspects of the work Xiaoxing Xiong: Acquisition or analysis or interpretation of data for the work; Drafting the work or revising it critically for important intellectual content; Final approval of the version to be published; Agreement to be accountable for all aspects of the work Chen Chen: Conception or design of the work; Acquisition or analysis or interpretation of data for the work; Drafting the work or revising it critically for important intellectual content; Final approval of the version to be published; Agreement to be accountable for all aspects of the work

Running Title: SGLT2i improves HPO axis disorder

Dual Publication: No

Funding: University of Queensland (UQ): Chen Chen, N/A; Australian NHMRC: Chen Chen, N/A Australian NHMRC and the University of Queensland.

SGLT2 inhibitor ameliorates high-fat diet-induced hypothalamic-pituitary-ovarian axis disorders

Xiaolin Chen^{1#}, Lili Huang^{2#}, Ling Cui³, Zhuoni Xiao⁴, Xiaoxing Xiong^{5*}, Chen Chen^{2*}

¹Department of Endocrinology, Renmin Hospital of Wuhan University, Wuhan, 430060, China.

²School of Biomedical Science, University of Queensland, St. Lucia, Brisbane, Queensland, 4072, Australia.

³Department of Reproduction and Infertility, Chengdu Women's and Children's Central Hospital, School of Medicine, University of Electronic Science and Technology of China, Chengdu, 611731, China.

⁴Reproductive Medical Center, Renmin Hospital of Wuhan University, Wuhan, 430060, China.

⁵Department of Neurosurgery, Renmin Hospital of Wuhan University, Wuhan, 430060, China.

[#]Xiaolin Chen and Lili Huang contributed equally to this work.

Correspondence:

*Chen Chen: chen.chen@uq.edu.au

*Xiaoxing Xiong : xiaoxingxiong@whu.edu.cn

Declaration of interest

No competing financial interests in this paper.

ABSTRACT

High-fat diet (HFD) consumption is known to be associated with ovulatory disorder among women of reproductive age. Previous studies in animal models suggested that HFD-induced microglia activation contributed to hypothalamus inflammation causing the dysfunction of the hypothalamic-pituitary-ovarian (HPO) axis, leading to subfertility. Sodium-glucose cotransporter 2 inhibitors (SGLT2i) are a novel class of lipid-soluble antidiabetic drugs to target primarily the early proximal tubules in the kidney. Recent evidence revealed an additional expression site of SGLT2 in the central nervous system (CNS), suggesting a promising site of action of SGLT2i in the CNS. In type 2 diabetes (T2D) patients and rodent models, SGLT2i exerted the neuroprotective properties through antioxidative stress, alleviation of cerebral atherosclerosis, and the suppression of microglia-induced neuroinflammation. Furthermore, clinical observations in patients with polycystic ovary syndrome (PCOS) demonstrated that SGLT-2i ameliorated the patient anthropometric parameters, body composition, and insulin resistance. Therefore, it is of importance to explore the CNS mechanism of SGLT-2i therapy in the recovery of reproductive function in patients with PCOS and obesity. Here, we review the hypothalamic inflammatory mechanisms of high-fat diet-induced microglial activation, with a focus on the clinical utility and possible mechanism of SGLT2i in promoting reproductive fitness.

Keywords: High-fat diet; Hypothalamic inflammation; HPO axis; SGLT2 inhibitor; Microglia;

Ovulatory disorder

Introduction

The consumption of a high-fat diet (HFD) in humans and animals increases the risk of developing cardiovascular diseases, type 2 diabetes, cognitive impairment, and also has negative impacts on reproductive fitness in both males and females (Chakraborty et al., 2016; Cheng et al., 2019; Holloway et al., 2011; McLean et al., 2019; Skaznik-Wikiel et al., 2016). Energy homeostasis is a key regulator of the reproductive system. Excessive energy consumption is closely related to hormonal disturbances and reproductive disorders. Polycystic ovary syndrome (PCOS) is a common reproductive endocrine-metabolic disease affecting 4-7% of women at reproductive age (Bozdag et al., 2016). PCOS women are inclined to consuming more HFD (Mizgier et al., 2021). Anovulation is the key characteristic of PCOS, which is often accompanied by insulin resistance, hyperandrogenism, hyperlipidemia, and abdominal obesity (Osibogun et al., 2020). Notwithstanding advancements in research regarding the mechanisms for such a tight relationship between energy excess and reproductive dysfunction, the underlying pathogenesis of the PCOS has not been elucidated. Increasing numbers of rodent models have provided increased comprehension of PCOS etiology (Patel and Shah, 2018; Roberts et al., 2017). However, there are no rodent models exhibiting all clinical traits of PCOS. Previous studies have shown that high-fat diets caused increased body fat content, insulin resistance, irregular estrous cycling, and ovulatory dysfunction in female mice (Lai et al., 2014; Wu et al., 2019). Feeding HFD from pre-pubertal age in female rats produced both reproductive and metabolic disturbances similar to the features of PCOS (Patel and Shah, 2018). Thus, the HFD-induced obesity mouse model has been widely used to study the underlying cause of PCOS. Although the exact pathophysiological mechanisms underlying the PCOS phenotypic characteristics remain largely unknown, a commonly accepted explanation is due to metabolic disturbance, often accompanied by insulin resistance. Therefore, it is no surprising to have

Metformin as the first line therapeutic medicine (Lovvik et al., 2019). However, as emerging novel drugs have been evidenced to improve insulin sensitivity and energy homeostasis in diabetic patients, we are in the position of revealing the potential of these drugs in treating PCOS.

SGLT2i are a new group of anti-hyperglycemia **drug** for treating type 2 diabetes. The mechanism of their action is to block glucose with SGLT2 receptors in the proximal tubules of the kidney and inhibiting urine glucose reabsorption. In addition to the direct value of SGLT2i in improving glucose homeostasis and insulin sensitivity in diabetes patients, many large double-blind clinical observations and meta-analysis also revealed the additional **benefit** effects in cardiovascular mortality, stroke events, and kidney protection in treatment of diabetic patients (Perkovic et al., 2019; Wiviott et al., 2019; Zinman et al., 2015.). Considering that the key metabolically disturbed trait of PCOS (defined as insulin resistance) resembled that of diabetes, and those accompanied alike features similar between the two (including , obesity, low grade inflammation, metabolic abnormalities, and increased risk of cardiovascular event) (Berni et al., 2021; Wekker et al., 2020), it is reasonable to propose that SGLT2i could be beneficial for patients with PCOS and it is worthwhile to explore any emerging mechanisms that are supplementary to improving metabolic abnormalities.

In this review, we summarize the mechanism of HFD-induced reproductive dysfunction, with a specific focus on hypothalamic inflammation and HPO axis disorder. **Meanwhile, we outline the clinical and experimental studies of SGLT2i in improving metabolic disorders and HPO axis dysfunction.**

Finally, we aim to explore the possible mechanism of SGLT2i in improving HPO axis dysfunction, and provide a theoretical basis for further research on promoting reproductive endocrine and metabolic diseases.

1. Consumption of HFD causes hypothalamic-pituitary-ovarian (HPO) axis dysfunction

Notwithstanding having limitations to fully represent the pathophysiology traits of PCOS in humans, animal models have boosted our understanding of the etiology of PCOS over decades beyond human studies. Among many animal models, the HFD induced obese mouse model is the most widely used one due to its abundant related traits as to PCOS, including increased body fat content, development of insulin resistance, irregular estrous cycling, and ovulatory dysfunction (Negron and Radovick, 2020; Ziarniak et al., 2018). HFD-related reproductive problems in female mice include early vaginal opening (VO), irregular estrous cycles, abnormal luteinizing hormone (LH) releasing, and compromise fertility due to anovulation (Mikhael et al., 2019). It is believed that HFD exposure induces reproductive dysfunction at multiple levels of the hypothalamic-pituitary-ovarian (HPO) axis leading toward ovulation disorders. Ovulation is triggered by LH surge, which is secondary to the surge-mode release of GnRH by the master neurons, i.e. GnRH neurons. Studies have evaluated the effects of HFD on GnRH secretion and reproductive function (Brothers et al., 2010; Chang et al., 2021; Hussain et al., 2016; Ma et al., 2016) and revealed that the GnRH pulse generator is extremely sensitive to energetic stress. In one study, **the literature** reported that HFD-induced obesity female DBA/2J mice manifested a more than 50% suppression of GnRH expression accompanied by more than a 60% decrease in natural pregnancy rates (Tortoriello et al., 2004)(**Fig. 1**). In another study, C57BL/6J female mice were placed on a 60% HFD for 32 weeks (Skaznik-Wikiel et al., 2016). Both obese mice and non-obese mice **were subfertility and show** impaired ovarian function regardless of the obese phenotype (Skaznik-Wikiel et al., 2016). This result suggests that HFD itself may cause the female reproductive dysfunction independent of the development of obesity. It is likely that HFD may act on the reproductive neurons to alter its secretion patterns as the pulsatile release of GnRH and LH is critical for follicle maturation in female mammals. This is supported

by a recent study which demonstrated that HFD increases LH pulse frequency at diestrus and **decreases** mean and basal LH levels at estrus in mice (Negron and Radovick, 2020)(**Fig. 1**).

In addition to the impact on the master neurons, HFD was also shown **detrimental** effects on ovaries and oocyte qualities. Compromised fertility and reduced primordial follicles in the ovary were observed in HFD mice, which was accompanied by high levels of pro-inflammatory cytokines and increased macrophage infiltration (Gao et al., 2021; Hohos et al., 2020; Skaznik-Wikiel et al., 2016). The ovaries from HFD-induced DIO mice showed more atresia follicles and matured follicles with lower quantities of granulosa cells (GCs) (Wen et al., 2020; Wu et al., 2017). The suggested mechanisms responsible for such ovarian dysfunction following long-term exposure to HFD include increased GCs apoptosis and the suppression of ovarian angiogenesis. GCs apoptosis may be largely attributed to the oxidative stress associated DNA damage and the endoplasmic reticulum (ER) stress induced by HFD (Wen et al., 2020; Wu et al., 2017)(**Fig. 1**). The suppressing ovarian angiogenesis, on the other hand, is considered as a consequence of the HFD induced inhibition of hypoxia inducible factor 1 α (HIF1 α)-vascular endothelial growth factor (VEGF) signaling, which subsequently arrests follicular development and formation of corpus luteum (CL) (Wu et al., 2017). In addition, other key ovarian genes are also dysregulated by HFD. For example, Hohos et al. have found that expression of endothelin-2 (Edn2) gene in ovary, a key gene for ovulation, is completely dysregulated across the estrous cycle in HFD-fed mice. Simultaneously, key ovarian steroidogenic genes, such as Star and Cyp19a1, are also dysregulated after HFD-exposed (Hohos et al., 2021).

Thus, HFD causes HPO axis dysfunction traversing multiple levels, from the central regulating neurons down to the peripheral ovarian glands. Inflammation induced by HFD is generally accepted as a

key molecule mechanism at levels of peripheral organs. It is yet to know whether HFD directly contributes to the hypothalamic inflammation that is supplemental to its effects on the master reproductive regulating neurons, hence, causes the dysfunction of the HPO axis

Fig. 1. HFD exposure induces hypothalamic- pituitary-ovarian (HPO) axis dysfunction leading to ovulation disorders. HFD may act on hypothalamic GnRH neurons and pituitary to alter the surge and pulsatile release of GnRH and LH, resulting in irregular estrous cycling and ovulation disorders. HFD may also act on ovary, inducing inflammation, endoplasmic reticulum (ER) stress and oxidative stress, leading to increased atresia follicles, granulosa cells (GCs) apoptosis and the suppression of ovarian angiogenesis. Black solid arrow, to stimulate; red blunt line, to inhibit.

2 HFD-induced hypothalamic inflammation is involved in HPO axis dysfunction

2.1 Microglia activation contributes to neuroinflammation

Microglia are the resident macrophages of the brain which require colony-stimulating factor 1 receptor (CSF1R) signaling for their survival (Alliot et al., 1999; Ginhoux et al., 2010). Representing 5–10% of the total cell population within the brain parenchyma (Aguzzi et al., 2013), microglia exert an important role in immune response and maintenance of tissue homeostasis by clearing pathogens, dying cells, debris or aberrant proteins (Faustino et al., 2011; Maas et al., 2020; Nugent et al., 2020; Villani et al., 2019). By responding to a wide variety of pathological stimuli, microglia scrutinize brain development, monitor synaptic function, regulate synaptogenesis, to protect the CNS under various pathological conditions throughout life (Hammond et al., 2019; Lund et al., 2018; Nguyen et al., 2020; Saitgareeva et al., 2020; Siew et al., 2019; Wake et al., 2009). With an extremely plastic phenotype that strongly **depend** on the local microenvironment, microglia shape the beneficial or deleterious functions in response to their environment change (Kalambogias et al., 2020; Klawonn et al., 2021; Pasciuto et al., 2020). Single-cell and single-nucleus RNAseq have revealed that each subtype microglia displays inherent properties and performs distinctive functions (Clark et al., 2021; Colonna and Brioschi, 2020; Lloyd et al., 2019). As such the numerous functions of microglia would be achieved through property of diverse phenotypes, each associated with unique molecular signatures (Song and Colonna, 2018). Although having a multi-functional potential, microglia are generally thought to benefit neurons when activated in physiological fitted conditions. However, under pathological conditions, activated microglia release inflammatory factors to induce neuroinflammation, which triggers neuronal death (Alexaki et al., 2018; Marschallinger et al., 2020; Ndoja et al., 2020). These proinflammatory microglia are characterized by ameboid morphology, high expression of ionized calcium-binding adaptor molecule 1 (IBA1), CD68, CD80, and CD11b (Merlini et al., 2019; Unger et al., 2018; Wachholz et al., 2016). The inflammatory cytokines include tumor necrosis factor- α (TNF- α), interleukin-1 β (IL-1 β), interleukin-18

(IL-18), superoxide, nitric oxide (NO), reactive oxygen species (ROS), and proteases (Lively and Schlichter, 2018; Plastira et al., 2017). In addition, the activation of microglia has demonstrated neuroprotection and **reparation**. These phenotype microglia exhibit anti-inflammatory action, high expression of IBA1 and CD206, and release of interleukin-10 (IL-10), IL-4, and TGF- β (Orihuela et al., 2016; Vergara et al., 2019)(Fig 2).

Fig. 2. Long chain saturated fatty acids (LCSFAs) induced activation of microglia contributing to neuroinflammation. Microglia polarize toward pro-inflammatory or anti-inflammatory phenotypes after exposure to HFD. Activated microglia, characterized by the expression of ionized calcium-binding adaptor molecule 1 (IBA1), CD68, CD80 and CD11b, release pro-inflammatory cytokines including tumor necrosis factor- α (TNF- α), interleukin-1 β (IL-1 β), interleukin-18 (IL-18), superoxide, nitric oxide (NO), reactive oxygen species (ROS), and proteases. Activated microglia, high expression of IBA1 and CD206, release anti-inflammatory cytokines including interleukin-10 (IL-10), IL-4, and TGF- β .

Many neurodegenerative diseases such as Alzheimer disease, Parkinson's Disease (PD), amyotrophic lateral sclerosis (ALS), and multiple sclerosis (MS), are linked with neuroinflammation

induced by activated microglia (Hickman et al., 2018). The NOD-, LRR- and pyrin domain-containing protein 3 (NLRP3) inflammasome activation in over-reactive microglia play a key role in the development and progression of neuroinflammation (Ising et al., 2019; Voet et al., 2018). The microglial NLRP3 inflammasome is multimeric complexes containing NLRP3, apoptosis-associated speck-like protein containing C-terminal caspase-activation and recruitment domain (CARD) (ASC), and pro-caspase 1. Upon activation, NLRP3 interaction with ASC through CARD-CARD result in recruiting pro-caspase 1, which then pro-caspase 1 is converted into caspase 1 (Mariathasan et al., 2004). Activated caspase 1 cleaves and activates pro- IL-1 β and pro- IL-18 into IL-1 β and IL-18. Finally, the activated microglia release IL-1 β and IL-18 contributing to development and progression of neuroinflammation and disease (Freeman et al., 2017; Voet et al., 2019)(Fig. 3).

2.2 High-fat diet induces pathological hypothalamic microglia activation

HFD enriched with long chain saturated fatty acids (LCSFAs) are known to directly impact on the central nervous system (D'Alonzo et al., 2020; Milanski et al., 2009; Valdearcos et al., 2014). Excessive intake of dietary LCSFAs increases brain-blood barrier (BBB) permeability and induces BBB dysfunction (D'Alonzo et al., 2020). With an anatomical proximity to BBB, the hypothalamus is the front-line brain region that would be initially affected by any detrimental factors that are leaking through the dysfunctional BBB (Thaler et al., 2012). Hypothalamus inflammation often occurs in animals fed on HFD and the transcription of inflammatory genes is activated as early as few hours after the first exposure to HFD (Thaler et al., 2012). In addition, neurogenesis is significantly attenuated in the hypothalamus of adult mice maintained on prolonged HFD feeding [16, 62], which could be secondary to HFD-induced neuroinflammatory responses. Evidences support a multi-factorial explanation of HFD

induced hypothalamic inflammatory , including infiltration of peripheral macrophages (Chen et al., 2021; Lainez et al., 2018), stimulation of resident microglia (Baufeld et al., 2016; Milanski et al., 2009; Zhang et al., 2008), and direct astrocyte activation. Among which, microglia have gained the most attention due to **the fact that** their local residential advantages.

Microglial inflammatory activation in the medio-basal hypothalamus (MBH) following HFD-feeding is associated with neuronal functional changes (Valdearcos et al., 2018; Valdearcos et al., 2014).

Researchers have shown that in HFD-induced hypothalamic inflammation, microglia change their appearance characterized by the prolonged processes emanating from a discretely enlarged cell body and molecular signature(Baufeld et al., 2016). Of interest, Tschop et al. **have** observed that microglial cells exert their function in a strict circadian pattern, with higher activity **occurs** during the dark, active phase, compared to the light, sleep phase under physiological conditions in mice(Yi et al., 2017). However, hypothalamic microglial cells from mice fed HFD lose their day/night rhythm and are persistently activated (Yi et al., 2017). This loss of daily rhythm of microglial circadian genes accompanied with changes in substrate utilization and energy production **that** may impair microglial immune-metabolic functions.

Therefore, HFD appears to have **impacts** on hypothalamic microglial cells at multiple levels, ranging from changes in phenotypic characteristics to functional properties. Whether these changes have crossed impacts on other hypothalamic neuronal populations that are responsible for growth (i.g. GHRH neurons) or reproductivity (i.g. GnRH neurons) require further investigations.

2.3 Possible signaling pathways of HFD-induced hypothalamic inflammation in microglia activation

So far, the molecular mechanisms underlying HFD-induced hypothalamic inflammation by abnormally activated microglia are not fully understood. Previous studies suggest that a Toll-like receptor 4 (TLR4)/ nuclear factor kappa-light-chain-enhancer of activated B cells (NF- κ B) signaling pathway in microglia injury (Muhammad et al., 2019). As receptors for lipopolysaccharide (LPS) and saturated and polyunsaturated fatty acids (SFAs), TLR4 has been shown to be expressed on the microglia membrane and mediates many neuroinflammatory diseases (Milanski et al., 2009; Nie et al., 2018). TLR4 loss-of-function mutation or pharmacological inhibition of TLR4 alleviates HFD-induced weight gain and attenuates HFD induced hypothalamic highly expression of inflammation factors such as IL-6 and IL-10 (Milanski et al., 2009). NF- κ B is a transcriptional factor complex located at the junction of the downstream signaling pathway of TLR4, the first-line molecules for initiating innate immune responses (Wang et al., 2012). The inhibition of TLR4/NF- κ B signaling pathway in microglia by means of pharmacological or antibody-based blocking agents improve HFD- induced neuroinflammatory (Jiao et al., 2018; Lee et al., 2012; Muhammad et al., 2019; Zhang et al., 2020; Zusso et al., 2019). Previous researches have revealed that reducing neuronal inflammatory capacity through inhibition of TLR4/NF- κ B pathway intermediates restores hypothalamic control of energy balance, resulting in reduced susceptibility to HFD-induced obesity (Benzler et al., 2015; Kleinridders et al., 2009; Zhang et al., 2008). Thus, it is likely that TLR4/ NF- κ B signaling may exert an important role in HFD induced microglia activation, contributing to neuronal inflammations (Fig. 3).

2.4. HFD induces shift in substrate utilization and microglial function

Consuming HFD disturbs microglial daily rhythmicity and stimulates microglial reactivity persistently in the MBH. The persistently activated microglia secrete tumor-necrosis factor- α (TNF α)

(Yi et al., 2017). TNF α stimulates induces mitochondrial stress in mouse primary hypothalamic neurons via stimulating mitochondrial ATP production. Moreover, TNF α induces mitochondrial elongation in neurites of the MBH neurons and drives neuronal energy demands by increasing neuronal firing rate (Yi et al., 2017). In long run, the persistently induced mitochondrial stress could contribute key neuronal dysfunction. It is well-known that daily rhythms generated in behavioral, physiological, and hormonal processes in mammals allow animals adapt to daily environmental changes, and optimizing metabolic function to the time of day. In the Yi et al. study, their results demonstrate that HFD leads to loss of daily rhythm of microglial circadian genes and impair microglial immunometabolic functions primarily at the transition period between dark and light phase, accompanied with changes in substrate utilization and energy production (Milanova et al., 2019). The shift in substrate utilization is known to have an effect on the activation status of immune cells. Immune cell function is highly dependent on metabolic adaptation of the immune cells, allowing for abrupt shifts in energy utilization, thus promoting either a resting or an activated state (Lee et al., 2018; O'Neill et al., 2016)(Fig. 3).

The microglial substrate utilization is focusing on glutamate, glucose and FA metabolism (Bernier et al., 2020; O'Neill et al., 2016). Yi et al. have shown a decrease in microglial glutamate utilization in the active period of HFD- fed animals during the dark phase as seen in glutamine conversion to glutamate and glutamate conversion to α -ketoglutarate (Milanova et al., 2019). Over above that, a similar change was observed for glucose metabolism with decreased glucose utilization during the dark phase during HFD. However, HFD leads to an increase in lipid utilization and sensing in microglia during the light phase, suggesting a shift to FA utilization during the sleep phase of the rat (Milanova et al., 2019). Additionally, HFD Increases mitochondrial bioenergetics and dynamics gene expression during the light

phase (Milanova et al., 2019)(**Fig. 3**). Finally, evaluation of monocyte functional gene expression **showed** small or absent effect of HFD on blood monocyte immunometabolism, suggesting that HFD specifically affects microglial immunometabolism in rats (Milanova et al., 2019). As a mitochondrial protein, uncoupling protein 2 (UCP2) has been proposed acting as a metabolic switcher in the substrate choice in the mitochondria. **Recent** study has shown that HFD promotes **increase** expression of UCP2 in hypothalamic microglia, which induces microglia activation and inflammation in the arcuate nucleus (Kim et al., 2019).

Fig. 3. The three possible signaling pathways of HFD-induced hypothalamic inflammation in microglia activation. A. The long chain saturated fatty acids (LCSFAs) in HFD activate microglia via the MyD88-dependent Toll-like receptor 4 (TLR4)/ nuclear factor kappa-light-chain-enhancer of activated B cells (NF-κB)/NLRP3 inflammasome signaling pathway. The activation of NF-κB translocate to the nucleus and code pro-interleukin-1β (IL-1β) and pro-interleukin-18 (IL-18). NLRP3

inflammasome interaction with ASC through CARD-CARD result in recruiting pro-caspase 1, **which** **then** pro-caspase 1 is converted into caspase 1. Activated caspase 1 cleaves and activates pro- IL-1 β and pro- IL-18 into IL-1 β and IL-18. **B.** The increased free fatty acid (FFA) binding to receptor-CD36 entering into microglia and translocating to the nucleus leads to loss of daily rhythm of microglial circadian genes, impair microglial immunometabolic functions and activate microglia. The persistently activated microglia secrete tumor-necrosis factor- α (TNF α). **C.** Meanwhile, the increased FFA enhances mitochondrial bioenergetics and dynamics gene expression during the light phase, resulting in decreased glucose utilization and increased lipid utilization, ultimately leading to production of a reactive oxygen species (ROS).

2.5. Hypothalamic inflammation may be an underlying pathophysiological mechanism of HFD-induced ovulation disorders

Hypothalamic GnRH neurons are key neurons that regulate reproductive system by generating pulsatile GnRH secretion. The majority of GnRH neurons are located in the arcuate nucleus (part of the medio-basal hypothalamus and in the medial preoptic area), and their pulse generating function **were** closely regulated by several other neurons in the hypothalamus. Kisspeptin neurons are mainly located in the ARC and the anteroventral periventricular nuclei (AVPV). Given the proximity kisspeptin is known to activate GnRH neurons for stimulating GnRH secretion (Rumpler et al., 2020). Neurons in the arcuate nucleus (ARC) that concomitantly express kisspeptin, neurokinin B (NKB) and dynorphin A are termed KNDy (Kisspeptin/NKB/Dyn) neurons (Bartzen-Sprauer et al., 2014; Goodman et al., 2013; Navarro et al., 2009). The KNDy neurons generate discrete neural signals and act in ACR to control activity of

GnRH/LH pulse generator, with NKB being a positive regulator of this circuitry, and Dyn being responsible for GnRH pulse termination (Mittelman-Smith et al., 2016; Nagae et al., 2021; Wakabayashi et al., 2010). Neuropeptide expression in KNDy neurons depends on the metabolic status (Ziarniak et al., 2020). Their activity is influenced by both negative (undernutrition) and positive (overnutrition) energy balance. Moreover, there is evidence that KNDy cells in the ARC are locally connected with AgRP/NPY and POMC/CART neurons (Backholer et al., 2010; True et al., 2013). These neuropeptides have impact on the hypothalamic GnRH pulse (Fig. 4).

Given the plastic property of KNDy neurons in response to metabolic challenges, it is of interest to know the influence of HFD on KNDy neuronal peptide expression in ARC. Yang et al. revealed HFD-fed was no effects in the arcuate *Tac2 mRNA* (which encodes NKB) expression in female mice. However, they reported a decrease in the expression of *Kiss1*, *Kiss1r*, and NKB receptor *mRNA (Tacr3)* in the ARC of HFD-fed females compared to control (Yang et al., 2016). Data from Ziarniak et al. demonstrate that HFD have no effects on the number of *Kiss1*, *Tacr3*, and *Dyn* immunoreactive neurons in the ARC of female mice (Ziarniak et al., 2018). Studies from other groups on *Dyn* expression in the ARC of HFD-fed female rodents corroborate this result (Minabe et al., 2021; Yang et al., 2016; Ziarniak et al., 2020).

Although inconclusive in terms of HFD on the neuron peptide expressions, it is generally accepted that KNDy neurons mediate estrogen negative feedback on LH secretion, relay progesterone (P) inhibition of pulsatile GnRH secretion and modulate the pulsatile release of GnRH whereas HFD can affect KNDy neuron activity and disrupt estrous cycle. Although it is still unclear what exact mechanisms that mediate the interaction between HFD and KNDy neurons, it is to some extent supports an inflammatory theory. It

has been noticed that hypothalamic inflammation occurs well before the development of weight gain and obesity following HFD consumption. This is thought to be independent of the obese phenotype as similar results were seen in the lean group. Hypothalamic inflammation induced by lipopolysaccharide (LPS) activated microglia which plays a vital role in the progression of hypothalamic inflammation (Zusso et al., 2019). Coincidentally, Fergani and colleagues found that administration of LPS decreased the percentage of activated kisspeptin and Dyn immunoreactive cells and then inhibited LH surge onset (Fergani et al., 2017). **This somehow, provide hints to form an HFD-hypothalamic inflammation** (microglia)-reproductive dysfunction (Kisspeptin/Dyn neurons) link (Fig. 4).

Clinically, many studies have identified that high carbohydrate diet and high saturated fat ingestion are associated with inflammation, insulin resistance, androgen excess, and high incidence of anovulation in women with PCOS (Gonzalez et al., 2020; Gonzalez et al., 2014). By contrast, anti-inflammatory dietary intervention contributes to improving insulin sensitivity, promoting physiological menstrual cycle and improving fertilization in PCOS women (Barrea et al., 2019; Salama et al., 2015). Thus, hypothalamic inflammation is likely to be an underlying mechanism **that** contributing to the development of PCOS and drugs able to improve inflammation could be a potential **therapeutic option.**

Fig. 4. Hypothalamic inflammation may be an underlying pathophysiological mechanism of HFD-induced HPO axis dysfunction. The kisspeptin neurons in the arcuate nucleus (ARC) that concomitantly express neurokinin B (NKB) and dynorphin A are termed KNDy (Kisspeptin/NKB/Dyn) neurons. NKB stimulates kisspeptin release via acting on KNDy neurons. Dyn inhibits kisspeptin release via KNDy neurons. The role of KNDy neurons are controlling gonadotropin-releasing hormone (GnRH)/luteinizing hormone (LH) pulse release, which is responsible for follicular development and estrogen negative feedback. The kisspeptin neurons located in the anteroventral periventricular nuclei (AVPV) in controlling GnRH/LH surge release is responsible for ovulation and estrogen positive feedback. The pro-inflammatory cytokines, such as TNF- α , IL-1 β , IL-18 and ROS, impair neurogenesis and increase neurodegeneration.

3 SGLT2i improve female reproductive function

SGLT2i **low** plasma glucose by inhibiting renal tubular absorption of glucose. The direct effect of SGLT2i on female reproductive functions has not been fully studied. In clinical practice, there are only a small number of studies that have evaluated the effect of SGLT2i in PCOS patients without diabetes. A 12-week, randomized open-label controlled trial was performed in obese women with PCOS. Participants were randomized to either empagliflozin at a dose of 25 mg daily or metformin 1500mg. The result demonstrated significant decreases in weight, body mass index (BMI), waist circumference (WC), hip circumference, basal metabolic rate, and fat mass in empagliflozin group compared with metformin group. But there were no significant changes in fasting glucose, insulin, androgen, and hs-CRP after empagliflozin treatment (Javed et al., 2019). In a recent randomized, double-blind, 2-week trial with a dual SGLT1/2i in PCOS patients showed that licogliflozin reduced hyperinsulinaemia by 70%, androstendione by 19%, and dehydroepiandrosteron sulphate (DHEAS) by 24% (Tan et al., 2021). In good time another study in obese women with PCOS has demonstrated that dapagliflozin 10mg daily for 24 weeks resulted in significant improvements in blood glucose, insulin sensitivity and secretion, and reductions in weight, abdominal adiposity, WC, BMI, testosterone and free androgen (Elkind-Hirsch et al., 2021). However, none these studies measured the effects of SGLT2i on the menstrual cycle, ovulatory disorder, or fertility in PCOS women.

In dihydrotestosterone-induced rat model of PCOS **research**, empagliflozin for 3 weeks decreased fat mass, plasma leptin, and blood pressure, but no improvement of hyperinsulinaemia, no impact on HbA1c and lipid profile (Pruett et al., 2021). In our preliminary study, we found that female melanocortin-4-receptor-deficient obese mice (MC4R KO) served a metabolic disorder similar to PCOS patients and demonstrated irregular estrous cycles and ovulation failure (Chen et al., 2017). Our

preliminary data **shown that** 14-week-old MC4R KO mice treated with dapagliflozin for 11 weeks **significant decreased body weight gain**, normalized estrous cycle, recovered pulsatile LH secretion, and restored ovulation (manuscript in preparation for publish). Meanwhile, expression of *Kiss1* mRNA and *Gnrh* mRNA in hypothalamus of MC4R KO mice was increased after dapagliflozin treatment **(manuscript in preparation for publish)**. Thus, we provide evidence that SGLT2i is able to improve ovulatory dysfunction, at least, in MC4R KO mice and this may be mediated via reviving the hypothalamic regulating neurons that are responsible for HPO axis.

4. The possible mechanism of SGLT2 inhibitor ameliorates HPO axis dysfunction

To **exert** the mechanism of SGLT2i on HPO axis, it is reasonable to inspect SGLT2 expressions. Though not found on ovaries, an increasing number of evidences have reported SGLT2 expression in central nervous system (CNS) such as the hippocampus, hypothalamus, and blood-brain barrier (BBB) endothelial cells (Chiba et al., 2020). SGLT2i are lipid-soluble drugs and can traverse the BBB and reach the CNS. This **was** supported by evidences demonstrating that SGLT2 inhibitors **neuroprotective** properties in patients and animals with type 2 diabetes (T2D), such as improving the impaired cognitive function (Hayden et al., 2019; Hierro-Bujalance et al., 2020; Lin et al., 2014) **and was further applied to** many neurodegenerative diseases, such as Alzheimer's Disease (AD), Parkinson's Disease (PD), Huntington's disease, epilepsy, and cerebral ischemia (Arab et al., 2021; El-Sahar et al., 2020; Hierro-Bujalance et al., 2020; Lin et al., 2014).

The neuroprotective effects of SGLT2i are likely to be via attenuating oxidative stress, reducing inflammation and improving immune response whereas microglia may play an important role. In *db/db*

T2D mouse, Lin B. et al report that empagliflozin treatment increases brain-derived neurotrophic factor (BDNF), the decline of which is associated with cognitive dysfunction (Franzmeier et al., 2021), and **attenuates** cerebral oxidative stress, which is association with cognitive impairment as well (Lin et al., 2014). A small number of researches have shown that cerebral atherosclerosis (CA) is also important contributor to **increase** prevalence of AD independently of ischemia (Bos et al., 2015; Wingo et al., 2020). **Studies in** experimental evidence **have** shown that SGLT2i alleviate atherosclerosis by reducing inflammation, improving endothelial function, inhibiting the activity of renin-angiotensin-aldosterone system, and modulating macrophage polarization (Day et al., 2020; Ganbaatar et al., 2020; Koyani et al., 2020; Liu et al., 2021; Nakatsu et al., 2017; Park et al., 2020). The bulk of evidence suggests that microglia-mediated neuroinflammation contributes to pathology and progression of AD (Santiago et al., 2021; Subbarayan et al., 2021). It has been shown that microglial activation induced by systemic inflammation via NLRP3 inflammasome impairs microglial clearance contributing to development and progression of AD (Tejera et al., 2019). The latest clinical research has demonstrated that treatment with empagliflozin inhibits NLRP3 inflammasome activity and IL-1 β **secreting** in diabetic patients at high risk of cardiovascular disease (Kim et al., 2020). HFD-induced inflammation in hypothalamus, as indicated by increased numbers of Iba1-positive microglia and IL6 mRNAs levels, was decreased in canagliflozin-treated mice (Naznin et al., 2017). Dapagliflozin also exerted anti-inflammatory effects in the brains via **inhibiting the** phosphorylation of NF- κ B and restored cognitive function in HFD-induced obese rats (Sa-Nguanmoo et al., 2017). In PD rat model, dapagliflozin **can reduces** oxidative stress of neuron by decreasing lipid peroxides, and consequently **restores** impaired DJ/Nrf2 pathway. **Meanwhile,** **it neutralizes** neuronal apoptosis induced by ROS via PI3K/AKT/GSK-3 β (Ser9) pathway. In addition, dapagliflozin inhibits the neuroinflammation via suppressing the NF- κ B signaling pathway activation

(Arab et al., 2021). Recently, a clinical study from Hong Kong reports that the use of SGLT2 inhibitor is associated with a decreased risk of PD (Mui et al., 2021).

These pieces of findings suggested that the SGLT2i prevented cognitive decline in those models related to the reduction in cerebral oxidative stress, alleviation of cerebral atherosclerosis, and suppression of microglia-mediated neuroinflammation. As many of the HFD induced pathological changes assembled to that of the neurodegenerative diseases, we speculate that the underlying mechanism for the beneficial effects of the SGLT2i on HFD-induced HPO axis disorders and/or PCOS may be similar to that of improving cognitive decline (Fig. 5). SGLT2i may rescue impaired GnRH surge in females and restore ovulatory function, as well as improve body weight and insulin resistance induced by HFD.

Fig. 5. The possible mechanism of SGLT2i ameliorates HPO axis dysfunction induced by HFD.

According to the evidences, SGLT2i have neuroprotective properties via the following pathways: A.

inhibiting microglial activation, suppression of the phosphorylation of NF-κB, and controlling NLRP3

inflammasome activity, that result in decreasing proinflammatory cytokines release and neuroinflammation. **B.** Increasing brain-derived neurotrophic factor (BDNF), decreasing lipid peroxides through restoring impaired DJ/Nrf2 pathway, that reduce oxidative stress of neuron. **C.** curbing the PI3K/AAKT//GSK-3 β (Ser9) pathway inhibit ROS-dependent neuronal apoptosis. Together, all these events ultimately ameliorate impaired neurogenesis and increased neurodegeneration. Black solid arrow, stimulate; red blunt line, inhibit; green solid arrow, stimulating effect of SGLT2i; green blunt line, inhibitory effect of SGLT2i.

5. Conclusions

HFD intake have detrimental effect on HPO axis function and female fertility that is independent of obese phenotype. Previous studies have shown that HFD activated microglia via TLR4/NF- κ B signaling pathway, leading to hypothalamic inflammation and impairment of KNDy neuron activity, consequently the disruption of oestrous cycle. However, the molecular mechanism of HFD-induced female reproductive dysfunctions remains largely unknown due to the limitations of relevant research. Inspired by mechanistic studies in in AD and PD patients by which SGLT2i have a positive impact in cognitive decline and neuronal apoptosis, we hypothesize that the possible underlying mechanisms underlying the beneficial effect of SGLT2i on HPO axis may be attributed to the reduction in brain oxidative stress and the amelioration of hypothalamic neuroinflammation. Based on the fact that HFD induces HPO axis dysfunction, this review, attempts to answer two of the questions: 1. Whether hypothalamic inflammation contributes to the etiology of infertility at hypothalamic levels? 2. Whether SGLT2i can be used as a potent therapy for PCOS and the relevant mechanisms. Although a full

landscape of the mechanism underlying HFD induced hypothalamic HPO axis dysfunction was limited due to lacking sufficient research studies, this review intends to start further discussion and attract more valuable research studies in future.

Reference

- Aguzzi, A., Barres, B.A., Bennett, M.L., 2013. Microglia: scapegoat, saboteur, or something else? *Science*. 339(6116), 156-161.
<https://doi.org/10.1126/science.1227901>.
- Alexaki, V.I., Fodelianaki, G., Neuwirth, A., Mund, C., Kourgiantaki, A., Ieronimaki, E., Lyroni, K., Troullinaki, M., Fujii, C., Kanczkowski, W., Ziogas, A., Peitzsch, M., Grossklaus, S., Sonnichsen, B., Gravanis, A., Bornstein, S.R., Charalampopoulos, I., Tsatsanis, C., Chavakis, T., 2018. DHEA inhibits acute microglia-mediated inflammation through activation of the TrkA-Akt1/2-CREB-Jmjd3 pathway. *Mol Psychiatry*. 23(6), 1410-1420.
<https://doi.org/10.1038/mp.2017.167>.
- Alliot, F., Godin, I., Pessac, B., 1999. Microglia derive from progenitors, originating from the yolk sac, and which proliferate in the brain. *Brain Res Dev Brain Res*. 117(2), 145-152. [https://doi.org/10.1016/s0165-3806\(99\)00113-3](https://doi.org/10.1016/s0165-3806(99)00113-3).
- Arab, H.H., Safar, M.M., Shahin, N.N., 2021. Targeting ROS-Dependent AKT/GSK-3beta/NF-kappaB and DJ-1/Nrf2 Pathways by Dapagliflozin Attenuates Neuronal Injury and Motor Dysfunction in Rotenone-Induced Parkinson's Disease Rat Model. *ACS Chem Neurosci*. 12(4), 689-703.
<https://doi.org/10.1021/acchemneuro.0c00722>.
- Backholer, K., Smith, J.T., Rao, A., Pereira, A., Iqbal, J., Ogawa, S., Li, Q., Clarke, I.J., 2010. Kisspeptin cells in the ewe brain respond to leptin and communicate with neuropeptide Y and proopiomelanocortin cells. *Endocrinology*. 151(5), 2233-2243. <https://doi.org/10.1210/en.2009-1190>.
- Barrea, L., Arnone, A., Annunziata, G., Muscogiuri, G., Laudisio, D., Salzano, C., Pugliese, G., Colao, A., Savastano, S., 2019. Adherence to the Mediterranean Diet, Dietary Patterns and Body Composition in Women with Polycystic Ovary Syndrome (PCOS). *Nutrients*. 11(10). <https://doi.org/10.3390/nu11102278>.
- Bartzen-Sprauer, J., Klosen, P., Ciofi, P., Mikkelsen, J.D., Simonneaux, V., 2014. Photoperiodic co-regulation of kisspeptin, neurokinin B and dynorphin in the hypothalamus of a seasonal rodent. *J Neuroendocrinol*. 26(8), 510-520.
<https://doi.org/10.1111/jne.12171>.
- Baufeld, C., Osterloh, A., Prokop, S., Miller, K.R., Heppner, F.L., 2016. High-fat

- diet-induced brain region-specific phenotypic spectrum of CNS resident microglia. *Acta Neuropathol.* 132(3), 361-375. <https://doi.org/10.1007/s00401-016-1595-4>.
- Benzler, J., Ganjam, G.K., Pretz, D., Oelkrug, R., Koch, C.E., Legler, K., Stohr, S., Culmsee, C., Williams, L.M., Tups, A., 2015. Central inhibition of IKKbeta/NF-kappaB signaling attenuates high-fat diet-induced obesity and glucose intolerance. *Diabetes.* 64(6), 2015-2027. <https://doi.org/10.2337/db14-0093>.
- Berni, T.R., Morgan, C.L., Rees, D.A., 2021. Women With Polycystic Ovary Syndrome Have an Increased Risk of Major Cardiovascular Events: a Population Study. *J Clin Endocrinol Metab.* 106(9), e3369-e3380. <https://doi.org/10.1210/clinem/dgab392>.
- Bernier, L.P., York, E.M., MacVicar, B.A., 2020. Immunometabolism in the Brain: How Metabolism Shapes Microglial Function. *Trends Neurosci.* 43(11), 854-869. <https://doi.org/10.1016/j.tins.2020.08.008>.
- Bos, D., Vernooij, M.W., de Bruijn, R.F., Koudstaal, P.J., Hofman, A., Franco, O.H., van der Lugt, A., Ikram, M.A., 2015. Atherosclerotic calcification is related to a higher risk of dementia and cognitive decline. *Alzheimers Dement.* 11(6), 639-647 e631. <https://doi.org/10.1016/j.jalz.2014.05.1758>.
- Bozdog, G., Mumusoglu, S., Zengin, D., Karabulut, E., Yildiz, B.O., 2016. The prevalence and phenotypic features of polycystic ovary syndrome: a systematic review and meta-analysis. *Hum Reprod.* 31(12), 2841-2855. <https://doi.org/10.1093/humrep/dew218>.
- Brothers, K.J., Wu, S., DiVall, S.A., Messmer, M.R., Kahn, C.R., Miller, R.S., Radovick, S., Wondisford, F.E., Wolfe, A., 2010. Rescue of obesity-induced infertility in female mice due to a pituitary-specific knockout of the insulin receptor. *Cell Metab.* 12(3), 295-305. <https://doi.org/10.1016/j.cmet.2010.06.010>.
- Chakraborty, T.R., Donthireddy, L., Adhikary, D., Chakraborty, S., 2016. Long-Term High Fat Diet Has a Profound Effect on Body Weight, Hormone Levels, and Estrous Cycle in Mice. *Med Sci Monit.* 22, 1601-1608. <https://doi.org/10.12659/msm.897628>.
- Chang, B., Song, C., Gao, H., Ma, T., Li, T., Ma, Q., Yao, T., Wang, M., Li, J., Yi, X., Tang, D., Cao, S., 2021. Leptin and inflammatory factors play a synergistic role in the regulation of reproduction in male mice through hypothalamic kisspeptin-mediated energy balance. *Reprod Biol Endocrinol.* 19(1), 12.

- <https://doi.org/10.1186/s12958-021-00698-0>.
- Chen, K.E., Lainez, N.M., Nair, M.G., Coss, D., 2021. Visceral adipose tissue imparts peripheral macrophage influx into the hypothalamus. *J Neuroinflammation*. 18(1), 140. <https://doi.org/10.1186/s12974-021-02183-2>.
- Chen, X., Huang, L., Tan, H.Y., Li, H., Wan, Y., Cowley, M., Veldhuis, J.D., Chen, C., 2017. Deficient melanocortin-4 receptor causes abnormal reproductive neuroendocrine profile in female mice. *Reproduction*. 153(3), 267-276. <https://doi.org/10.1530/REP-16-0341>.
- Cheng, Y., Yu, X., Zhang, J., Chang, Y., Xue, M., Li, X., Lu, Y., Li, T., Meng, Z., Su, L., Sun, B., Chen, L., 2019. Pancreatic kallikrein protects against diabetic retinopathy in KK Cg-A(y)/J and high-fat diet/streptozotocin-induced mouse models of type 2 diabetes. *Diabetologia*. 62(6), 1074-1086. <https://doi.org/10.1007/s00125-019-4838-9>.
- Chiba, Y., Sugiyama, Y., Nishi, N., Nonaka, W., Murakami, R., Ueno, M., 2020. Sodium/glucose cotransporter 2 is expressed in choroid plexus epithelial cells and ependymal cells in human and mouse brains. *Neuropathology*. 40(5), 482-491. <https://doi.org/10.1111/neup.12665>.
- Clark, I.C., Gutierrez-Vazquez, C., Wheeler, M.A., Li, Z., Rothhammer, V., Linnerbauer, M., Sanmarco, L.M., Guo, L., Blain, M., Zandee, S.E.J., Chao, C.C., Batterman, K.V., Schwabenland, M., Lotfy, P., Tejeda-Velarde, A., Hewson, P., Manganeli Polonio, C., Shultis, M.W., Salem, Y., Tjon, E.C., Fonseca-Castro, P.H., Borucki, D.M., Alves de Lima, K., Plasencia, A., Abate, A.R., Rosene, D.L., Hodgetts, K.J., Prinz, M., Antel, J.P., Prat, A., Quintana, F.J., 2021. Barcoded viral tracing of single-cell interactions in central nervous system inflammation. *Science*. 372(6540). <https://doi.org/10.1126/science.abf1230>.
- Colonna, M., Brioschi, S., 2020. Neuroinflammation and neurodegeneration in human brain at single-cell resolution. *Nat Rev Immunol*. 20(2), 81-82. <https://doi.org/10.1038/s41577-019-0262-0>.
- D'Alonzo, Z., Lam, V., Nesbit, M., Graneri, L., Takechi, R., Mamo, J.C.L., 2020. Chronic Consumption of Bovine Dairy Milk Attenuates Dietary Saturated Fatty Acid-Induced Blood-Brain Barrier Dysfunction. *Front Nutr*. 7, 58. <https://doi.org/10.3389/fnut.2020.00058>.
- Day, E.A., Ford, R.J., Lu, J.H., Lu, R., Lundenberg, L., Desjardins, E.M., Green, A.E.,

- Lally, J.S.V., Schertzer, J.D., Steinberg, G.R., 2020. The SGLT2 inhibitor canagliflozin suppresses lipid synthesis and interleukin-1 beta in ApoE deficient mice. *Biochem J.* 477(12), 2347-2361. <https://doi.org/10.1042/BCJ20200278>.
- El-Sahar, A.E., Rastanawi, A.A., El-Yamany, M.F., Saad, M.A., 2020. Dapagliflozin improves behavioral dysfunction of Huntington's disease in rats via inhibiting apoptosis-related glycolysis. *Life Sci.* 257, 118076. <https://doi.org/10.1016/j.lfs.2020.118076>.
- Elkind-Hirsch, K.E., Chappell, N., Seidemann, E., Storment, J., Bellanger, D., 2021. Exenatide, Dapagliflozin, or Phentermine/Topiramate Differentially Affect Metabolic Profiles in Polycystic Ovary Syndrome. *J Clin Endocrinol Metab.* 106(10), 3019-3033. <https://doi.org/10.1210/clinem/dgab408>.
- Faustino, J.V., Wang, X., Johnson, C.E., Klibanov, A., Derugin, N., Wendland, M.F., Vexler, Z.S., 2011. Microglial cells contribute to endogenous brain defenses after acute neonatal focal stroke. *J Neurosci.* 31(36), 12992-13001. <https://doi.org/10.1523/JNEUROSCI.2102-11.2011>.
- Fergani, C., Routly, J.E., Jones, D.N., Pickavance, L.C., Smith, R.F., Dobson, H., 2017. KNDy neurone activation prior to the LH surge of the ewe is disrupted by LPS. *Reproduction.* 154(3), 281-292. <https://doi.org/10.1530/REP-17-0191>.
- Franzmeier, N., Ren, J., Damm, A., Monte-Rubio, G., Boada, M., Ruiz, A., Ramirez, A., Jessen, F., Duzel, E., Rodriguez Gomez, O., Benzinger, T., Goate, A., Karch, C.M., Fagan, A.M., McDade, E., Buerger, K., Levin, J., Duering, M., Dichgans, M., Suarez-Calvet, M., Haass, C., Gordon, B.A., Lim, Y.Y., Masters, C.L., Janowitz, D., Catak, C., Wolfsgruber, S., Wagner, M., Milz, E., Moreno-Grau, S., Teipel, S., Grothe, M.J., Kilimann, I., Rossor, M., Fox, N., Laske, C., Chhatwal, J., Falkai, P., Perneczky, R., Lee, J.H., Spottke, A., Boecker, H., Brosseron, F., Fliessbach, K., Heneka, M.T., Nestor, P., Peters, O., Fuentes, M., Menne, F., Priller, J., Spruth, E.J., Franke, C., Schneider, A., Westerteicher, C., Speck, O., Wiltfang, J., Bartels, C., Araque Caballero, M.A., Metzger, C., Bittner, D., Salloway, S., Danek, A., Hassenstab, J., Yakushev, I., Schofield, P.R., Morris, J.C., Bateman, R.J., Ewers, M., 2021. The BDNFVal66Met SNP modulates the association between beta-amyloid and hippocampal disconnection in Alzheimer's disease. *Mol Psychiatry.* 26(2), 614-628. <https://doi.org/10.1038/s41380-019-0404-6>.
- Freeman, L., Guo, H., David, C.N., Brickey, W.J., Jha, S., Ting, J.P., 2017. NLR

- members NLRC4 and NLRP3 mediate sterile inflammasome activation in microglia and astrocytes. *J Exp Med.* 214(5), 1351-1370.
<https://doi.org/10.1084/jem.20150237>.
- Ganbaatar, B., Fukuda, D., Shinohara, M., Yagi, S., Kusunose, K., Yamada, H., Soeki, T., Hirata, K.I., Sata, M., 2020. Empagliflozin ameliorates endothelial dysfunction and suppresses atherogenesis in diabetic apolipoprotein E-deficient mice. *Eur J Pharmacol.* 875, 173040. <https://doi.org/10.1016/j.ejphar.2020.173040>.
- Gao, X., Li, Y., Ma, Z., Jing, J., Zhang, Z., Liu, Y., Ding, Z., 2021. Obesity induces morphological and functional changes in female reproductive system through increases in NF-kappaB and MAPK signaling in mice. *Reprod Biol Endocrinol.* 19(1), 148. <https://doi.org/10.1186/s12958-021-00833-x>.
- Ginhoux, F., Greter, M., Leboeuf, M., Nandi, S., See, P., Gokhan, S., Mehler, M.F., Conway, S.J., Ng, L.G., Stanley, E.R., Samokhvalov, I.M., Merad, M., 2010. Fate mapping analysis reveals that adult microglia derive from primitive macrophages. *Science.* 330(6005), 841-845. <https://doi.org/10.1126/science.1194637>.
- Gonzalez, F., Considine, R.V., Abdelhadi, O.A., Acton, A.J., 2020. Inflammation Triggered by Saturated Fat Ingestion Is Linked to Insulin Resistance and Hyperandrogenism in Polycystic Ovary Syndrome. *J Clin Endocrinol Metab.* 105(6). <https://doi.org/10.1210/clinem/dgaa108>.
- Gonzalez, F., Sia, C.L., Shepard, M.K., Rote, N.S., Minium, J., 2014. The altered mononuclear cell-derived cytokine response to glucose ingestion is not regulated by excess adiposity in polycystic ovary syndrome. *J Clin Endocrinol Metab.* 99(11), E2244-2251. <https://doi.org/10.1210/jc.2014-2046>.
- Goodman, R.L., Hileman, S.M., Nestor, C.C., Porter, K.L., Connors, J.M., Hardy, S.L., Millar, R.P., Cernea, M., Coolen, L.M., Lehman, M.N., 2013. Kisspeptin, neurokinin B, and dynorphin act in the arcuate nucleus to control activity of the GnRH pulse generator in ewes. *Endocrinology.* 154(11), 4259-4269.
<https://doi.org/10.1210/en.2013-1331>.
- Hammond, T.R., Dufort, C., Dissing-Olesen, L., Giera, S., Young, A., Wysoker, A., Walker, A.J., Gergits, F., Segel, M., Nemes, J., Marsh, S.E., Saunders, A., Macosko, E., Ginhoux, F., Chen, J., Franklin, R.J.M., Piao, X., McCarroll, S.A., Stevens, B., 2019. Single-Cell RNA Sequencing of Microglia throughout the Mouse Lifespan and in the Injured Brain Reveals Complex Cell-State Changes.

- Immunity. 50(1), 253-271 e256. <https://doi.org/10.1016/j.immuni.2018.11.004>.
- Hayden, M.R., Grant, D.G., Aroor, A.R., DeMarco, V.G., 2019. Empagliflozin Ameliorates Type 2 Diabetes-Induced Ultrastructural Remodeling of the Neurovascular Unit and Neuroglia in the Female db/db Mouse. *Brain Sci.* 9(3). <https://doi.org/10.3390/brainsci9030057>.
- Hickman, S., Izzy, S., Sen, P., Morsett, L., El Khoury, J., 2018. Microglia in neurodegeneration. *Nat Neurosci.* 21(10), 1359-1369. <https://doi.org/10.1038/s41593-018-0242-x>.
- Hierro-Bujalance, C., Infante-Garcia, C., Del Marco, A., Herrera, M., Carranza-Naval, M.J., Suarez, J., Alves-Martinez, P., Lubian-Lopez, S., Garcia-Alloza, M., 2020. Empagliflozin reduces vascular damage and cognitive impairment in a mixed murine model of Alzheimer's disease and type 2 diabetes. *Alzheimers Res Ther.* 12(1), 40. <https://doi.org/10.1186/s13195-020-00607-4>.
- Hohos, N.M., Elliott, E.M., Cho, K.J., Lin, I.S., Rudolph, M.C., Skaznik-Wikiel, M.E., 2020. High-fat diet-induced dysregulation of ovarian gene expression is restored with chronic omega-3 fatty acid supplementation. *Mol Cell Endocrinol.* 499, 110615. <https://doi.org/10.1016/j.mce.2019.110615>.
- Hohos, N.M., Elliott, E.M., Giornazi, A., Silva, E., Rice, J.D., Skaznik-Wikiel, M.E., 2021. High-fat diet induces an ovulatory defect associated with dysregulated endothelin-2 in mice. *Reproduction.* 161(3), 307-317. <https://doi.org/10.1530/REP-20-0290>.
- Holloway, C.J., Cochlin, L.E., Emmanuel, Y., Murray, A., Codreanu, I., Edwards, L.M., Szmigielski, C., Tyler, D.J., Knight, N.S., Saxby, B.K., Lambert, B., Thompson, C., Neubauer, S., Clarke, K., 2011. A high-fat diet impairs cardiac high-energy phosphate metabolism and cognitive function in healthy human subjects. *Am J Clin Nutr.* 93(4), 748-755. <https://doi.org/10.3945/ajcn.110.002758>.
- Hussain, M.A., Abogresha, N.M., Hassan, R., Tamany, D.A., Lotfy, M., 2016. Effect of feeding a high-fat diet independently of caloric intake on reproductive function in diet-induced obese female rats. *Arch Med Sci.* 12(4), 906-914. <https://doi.org/10.5114/aoms.2016.59790>.
- Ising, C., Venegas, C., Zhang, S., Scheiblich, H., Schmidt, S.V., Vieira-Saecker, A., Schwartz, S., Albasset, S., McManus, R.M., Tejera, D., Griep, A., Santarelli, F., Brosseron, F., Opitz, S., Stunden, J., Merten, M., Kayed, R., Golenbock, D.T.,

- Blum, D., Latz, E., Buee, L., Heneka, M.T., 2019. NLRP3 inflammasome activation drives tau pathology. *Nature*. 575(7784), 669-673.
<https://doi.org/10.1038/s41586-019-1769-z>.
- Javed, Z., Papageorgiou, M., Deshmukh, H., Rigby, A.S., Qamar, U., Abbas, J., Khan, A.Y., Kilpatrick, E.S., Atkin, S.L., Sathyapalan, T., 2019. Effects of empagliflozin on metabolic parameters in polycystic ovary syndrome: A randomized controlled study. *Clin Endocrinol (Oxf)*. 90(6), 805-813. <https://doi.org/10.1111/cen.13968>.
- Jiao, F.Z., Wang, Y., Zhang, H.Y., Zhang, W.B., Wang, L.W., Gong, Z.J., 2018. Histone Deacetylase 2 Inhibitor CAY10683 Alleviates Lipopolysaccharide Induced Neuroinflammation Through Attenuating TLR4/NF-kappaB Signaling Pathway. *Neurochem Res*. 43(6), 1161-1170. <https://doi.org/10.1007/s11064-018-2532-9>.
- Kalambogias, J., Chen, C.C., Khan, S., Son, T., Werberger, R., Headlam, C., Lin, C., Brumberg, J.C., 2020. Development and sensory experience dependent regulation of microglia in barrel cortex. *J Comp Neurol*. 528(4), 559-573.
<https://doi.org/10.1002/cne.24771>.
- Kim, J.D., Yoon, N.A., Jin, S., Diano, S., 2019. Microglial UCP2 Mediates Inflammation and Obesity Induced by High-Fat Feeding. *Cell Metab*. 30(5), 952-962 e955. <https://doi.org/10.1016/j.cmet.2019.08.010>.
- Kim, S.R., Lee, S.G., Kim, S.H., Kim, J.H., Choi, E., Cho, W., Rim, J.H., Hwang, I., Lee, C.J., Lee, M., Oh, C.M., Jeon, J.Y., Gee, H.Y., Kim, J.H., Lee, B.W., Kang, E.S., Cha, B.S., Lee, M.S., Yu, J.W., Cho, J.W., Kim, J.S., Lee, Y.H., 2020. SGLT2 inhibition modulates NLRP3 inflammasome activity via ketones and insulin in diabetes with cardiovascular disease. *Nat Commun*. 11(1), 2127.
<https://doi.org/10.1038/s41467-020-15983-6>.
- Klawonn, A.M., Fritz, M., Castany, S., Pignatelli, M., Canal, C., Simila, F., Tejada, H.A., Levinsson, J., Jaarola, M., Jakobsson, J., Hidalgo, J., Heilig, M., Bonci, A., Engblom, D., 2021. Microglial activation elicits a negative affective state through prostaglandin-mediated modulation of striatal neurons. *Immunity*. 54(2), 225-234 e226. <https://doi.org/10.1016/j.immuni.2020.12.016>.
- Kleinridders, A., Schenten, D., Konner, A.C., Belgardt, B.F., Mauer, J., Okamura, T., Wunderlich, F.T., Medzhitov, R., Bruning, J.C., 2009. MyD88 signaling in the CNS is required for development of fatty acid-induced leptin resistance and diet-induced obesity. *Cell Metab*. 10(4), 249-259.

- <https://doi.org/10.1016/j.cmet.2009.08.013>.
- Koyani, C.N., Plastira, I., Sourij, H., Hallstrom, S., Schmidt, A., Rainer, P.P., Bugger, H., Frank, S., Malle, E., von Lewinski, D., 2020. Empagliflozin protects heart from inflammation and energy depletion via AMPK activation. *Pharmacol Res.* 158, 104870. <https://doi.org/10.1016/j.phrs.2020.104870>.
- Lai, H., Jia, X., Yu, Q., Zhang, C., Qiao, J., Guan, Y., Kang, J., 2014. High-fat diet induces significant metabolic disorders in a mouse model of polycystic ovary syndrome. *Biol Reprod.* 91(5), 127. <https://doi.org/10.1095/biolreprod.114.120063>.
- Lainez, N.M., Jonak, C.R., Nair, M.G., Ethell, I.M., Wilson, E.H., Carson, M.J., Coss, D., 2018. Diet-Induced Obesity Elicits Macrophage Infiltration and Reduction in Spine Density in the Hypothalamus of Male but Not Female Mice. *Front Immunol.* 9, 1992. <https://doi.org/10.3389/fimmu.2018.01992>.
- Lee, Y.H., Jeon, S.H., Kim, S.H., Kim, C., Lee, S.J., Koh, D., Lim, Y., Ha, K., Shin, S.Y., 2012. A new synthetic chalcone derivative, 2-hydroxy-3',5,5'-trimethoxychalcone (DK-139), suppresses the Toll-like receptor 4-mediated inflammatory response through inhibition of the Akt/NF-kappaB pathway in BV2 microglial cells. *Exp Mol Med.* 44(6), 369-377. <https://doi.org/10.3858/emm.2012.44.6.042>.
- Lee, Y.S., Wollam, J., Olefsky, J.M., 2018. An Integrated View of Immunometabolism. *Cell.* 172(1-2), 22-40. <https://doi.org/10.1016/j.cell.2017.12.025>.
- Lin, B., Koibuchi, N., Hasegawa, Y., Sueta, D., Toyama, K., Uekawa, K., Ma, M., Nakagawa, T., Kusaka, H., Kim-Mitsuyama, S., 2014. Glycemic control with empagliflozin, a novel selective SGLT2 inhibitor, ameliorates cardiovascular injury and cognitive dysfunction in obese and type 2 diabetic mice. *Cardiovasc Diabetol.* 13, 148. <https://doi.org/10.1186/s12933-014-0148-1>.
- Liu, Y., Xu, J., Wu, M., Xu, B., Kang, L., 2021. Empagliflozin protects against atherosclerosis progression by modulating lipid profiles and sympathetic activity. *Lipids Health Dis.* 20(1), 5. <https://doi.org/10.1186/s12944-021-01430-y>.
- Lively, S., Schlichter, L.C., 2018. Microglia Responses to Pro-inflammatory Stimuli (LPS, IFN γ +TNF α) and Reprogramming by Resolving Cytokines (IL-4, IL-10). *Front Cell Neurosci.* 12, 215. <https://doi.org/10.3389/fncel.2018.00215>.
- Lloyd, A.F., Davies, C.L., Holloway, R.K., Labrak, Y., Ireland, G., Carradori, D., Dillenburg, A., Borger, E., Soong, D., Richardson, J.C., Kuhlmann, T., Williams,

- A., Pollard, J.W., des Rieux, A., Priller, J., Miron, V.E., 2019. Central nervous system regeneration is driven by microglia necroptosis and repopulation. *Nat Neurosci.* 22(7), 1046-1052. <https://doi.org/10.1038/s41593-019-0418-z>.
- Lovvik, T.S., Carlsen, S.M., Salvesen, O., Steffensen, B., Bixo, M., Gomez-Real, F., Lonnebotn, M., Hestvold, K.V., Zabielska, R., Hirschberg, A.L., Trouva, A., Thorarinsdottir, S., Hjelle, S., Berg, A.H., Andrae, F., Poromaa, I.S., Mohlin, J., Underdal, M., Vanky, E., 2019. Use of metformin to treat pregnant women with polycystic ovary syndrome (PregMet2): a randomised, double-blind, placebo-controlled trial. *Lancet Diabetes Endocrinol.* 7(4), 256-266. [https://doi.org/10.1016/S2213-8587\(19\)30002-6](https://doi.org/10.1016/S2213-8587(19)30002-6).
- Lund, H., Pieber, M., Parsa, R., Grommisch, D., Ewing, E., Kular, L., Han, J., Zhu, K., Nijssen, J., Hedlund, E., Needhamsen, M., Ruhrmann, S., Guerreiro-Cacais, A.O., Berglund, R., Forteza, M.J., Ketelhuth, D.F.J., Butovsky, O., Jagodic, M., Zhang, X.M., Harris, R.A., 2018. Fatal demyelinating disease is induced by monocyte-derived macrophages in the absence of TGF-beta signaling. *Nat Immunol.* 19(5), 1-7. <https://doi.org/10.1038/s41590-018-0091-5>.
- Ma, X., Hayes, E., Prizant, H., Srivastava, R.K., Hammes, S.R., Sen, A., 2016. Leptin-Induced CART (Cocaine- and Amphetamine-Regulated Transcript) Is a Novel Intraovarian Mediator of Obesity-Related Infertility in Females. *Endocrinology.* 157(3), 1248-1257. <https://doi.org/10.1210/en.2015-1750>.
- Maas, S.L.N., Abels, E.R., Van De Haar, L.L., Zhang, X., Morsett, L., Sil, S., Guedes, J., Sen, P., Prabhakar, S., Hickman, S.E., Lai, C.P., Ting, D.T., Breakefield, X.O., Broekman, M.L.D., El Khoury, J., 2020. Glioblastoma hijacks microglial gene expression to support tumor growth. *J Neuroinflammation.* 17(1), 120. <https://doi.org/10.1186/s12974-020-01797-2>.
- Mariathasan, S., Newton, K., Monack, D.M., Vucic, D., French, D.M., Lee, W.P., Roose-Girma, M., Erickson, S., Dixit, V.M., 2004. Differential activation of the inflammasome by caspase-1 adaptors ASC and Ipaf. *Nature.* 430(6996), 213-218. <https://doi.org/10.1038/nature02664>.
- Marschallinger, J., Iram, T., Zardeneta, M., Lee, S.E., Lehallier, B., Haney, M.S., Pluvinaige, J.V., Mathur, V., Hahn, O., Morgens, D.W., Kim, J., Tevini, J., Felder, T.K., Wolinski, H., Bertozzi, C.R., Bassik, M.C., Aigner, L., Wyss-Coray, T., 2020. Lipid-droplet-accumulating microglia represent a dysfunctional and

- proinflammatory state in the aging brain. *Nat Neurosci.* 23(2), 194-208.
<https://doi.org/10.1038/s41593-019-0566-1>.
- McLean, F.H., Campbell, F.M., Sergi, D., Grant, C., Morris, A.C., Hay, E.A., MacKenzie, A., Mayer, C.D., Langston, R.F., Williams, L.M., 2019. Early and reversible changes to the hippocampal proteome in mice on a high-fat diet. *Nutr Metab (Lond)*. 16, 57. <https://doi.org/10.1186/s12986-019-0387-y>.
- Merlini, M., Rafalski, V.A., Rios Coronado, P.E., Gill, T.M., Ellisman, M., Muthukumar, G., Subramanian, K.S., Ryu, J.K., Syme, C.A., Davalos, D., Seeley, W.W., Mucke, L., Nelson, R.B., Akassoglou, K., 2019. Fibrinogen Induces Microglia-Mediated Spine Elimination and Cognitive Impairment in an Alzheimer's Disease Model. *Neuron*. 101(6), 1099-1108 e1096.
<https://doi.org/10.1016/j.neuron.2019.01.014>.
- Mikhael, S., Punjala-Patel, A., Gavrilova-Jordan, L., 2019. Hypothalamic-Pituitary-Ovarian Axis Disorders Impacting Female Fertility. *Biomedicines*. 7(1). <https://doi.org/10.3390/biomedicines7010005>.
- Milanova, I.V., Kalsbeek, M.J.T., Wang, X.L., Korpel, N.L., Stenvers, D.J., Wolff, S.E.C., de Goede, P., Heijboer, A.C., Fliers, E., la Fleur, S.E., Kalsbeek, A., Yi, C.X., 2019. Diet-Induced Obesity Disturbs Microglial Immunometabolism in a Time-of-Day Manner. *Front Endocrinol (Lausanne)*. 10, 424.
<https://doi.org/10.3389/fendo.2019.00424>.
- Milanski, M., Degasperi, G., Coope, A., Morari, J., Denis, R., Cintra, D.E., Tsukumo, D.M., Anhe, G., Amaral, M.E., Takahashi, H.K., Curi, R., Oliveira, H.C., Carvalheira, J.B., Bordin, S., Saad, M.J., Velloso, L.A., 2009. Saturated fatty acids produce an inflammatory response predominantly through the activation of TLR4 signaling in hypothalamus: implications for the pathogenesis of obesity. *J Neurosci*. 29(2), 359-370. <https://doi.org/10.1523/JNEUROSCI.2760-08.2009>.
- Minabe, S., Iwata, K., Tsuchida, H., Tsukamura, H., Ozawa, H., 2021. Effect of diet-induced obesity on kisspeptin-neurokinin B-dynorphin A neurons in the arcuate nucleus and luteinizing hormone secretion in sex hormone-primed male and female rats. *Peptides*. 142, 170546.
<https://doi.org/10.1016/j.peptides.2021.170546>.
- Mittelman-Smith, M.A., Krajewski-Hall, S.J., McMullen, N.T., Rance, N.E., 2016. Ablation of KNDy Neurons Results in Hypogonadotropic Hypogonadism and

- Amplifies the Steroid-Induced LH Surge in Female Rats. *Endocrinology*. 157(5), 2015-2027. <https://doi.org/10.1210/en.2015-1740>.
- Mizgier, M., Jarzabek-Bielecka, G., Formanowicz, D., Jodlowska-Siewert, E., Mruczyk, K., Cisek-Wozniak, A., Kedzia, W., Opydo-Szymaczek, J., 2021. Dietary and Physical Activity Habits in Adolescent Girls with Polycystic Ovary Syndrome (PCOS)-HAstudy. *J Clin Med*. 10(16). <https://doi.org/10.3390/jcm10163469>.
- Muhammad, T., Ikram, M., Ullah, R., Rehman, S.U., Kim, M.O., 2019. Hesperetin, a Citrus Flavonoid, Attenuates LPS-Induced Neuroinflammation, Apoptosis and Memory Impairments by Modulating TLR4/NF-kappaB Signaling. *Nutrients*. 11(3). <https://doi.org/10.3390/nu11030648>.
- Mui, J.V., Zhou, J., Lee, S., Leung, K.S.K., Lee, T.T.L., Chou, O.H.I., Tsang, S.L., Wai, A.K.C., Liu, T., Wong, W.T., Chang, C., Tse, G., Zhang, Q., 2021. Sodium-Glucose Cotransporter 2 (SGLT2) Inhibitors vs. Dipeptidyl Peptidase-4 (DPP4) Inhibitors for New-Onset Dementia: A Propensity Score-Matched Population-Based Study With Competing Risk Analysis. *Front Cardiovasc Med*. 8, 747620. <https://doi.org/10.3389/fcvm.2021.747620>.
- Nagae, M., Uenoyama, Y., Okamoto, S., Tsuchida, H., Ikegami, K., Goto, T., Majarune, S., Nakamura, S., Sanbo, M., Hirabayashi, M., Kobayashi, K., Inoue, N., Tsukamura, H., 2021. Direct evidence that KNDy neurons maintain gonadotropin pulses and folliculogenesis as the GnRH pulse generator. *Proc Natl Acad Sci U S A*. 118(5). <https://doi.org/10.1073/pnas.2009156118>.
- Nakatsu, Y., Kokubo, H., Bumdelger, B., Yoshizumi, M., Yamamotoya, T., Matsunaga, Y., Ueda, K., Inoue, Y., Inoue, M.K., Fujishiro, M., Kushiya, A., Ono, H., Sakoda, H., Asano, T., 2017. The SGLT2 Inhibitor Luseogliflozin Rapidly Normalizes Aortic mRNA Levels of Inflammation-Related but Not Lipid-Metabolism-Related Genes and Suppresses Atherosclerosis in Diabetic ApoE KO Mice. *Int J Mol Sci*. 18(8). <https://doi.org/10.3390/ijms18081704>.
- Navarro, V.M., Gottsch, M.L., Chavkin, C., Okamura, H., Clifton, D.K., Steiner, R.A., 2009. Regulation of gonadotropin-releasing hormone secretion by kisspeptin/dynorphin/neurokinin B neurons in the arcuate nucleus of the mouse. *J Neurosci*. 29(38), 11859-11866. <https://doi.org/10.1523/JNEUROSCI.1569-09.2009>.
- Naznin, F., Sakoda, H., Okada, T., Tsubouchi, H., Waise, T.M., Arakawa, K.,

- Nakazato, M., 2017. Canagliflozin, a sodium glucose cotransporter 2 inhibitor, attenuates obesity-induced inflammation in the nodose ganglion, hypothalamus, and skeletal muscle of mice. *Eur J Pharmacol.* 794, 37-44.
<https://doi.org/10.1016/j.ejphar.2016.11.028>.
- Ndoja, A., Reja, R., Lee, S.H., Webster, J.D., Ngu, H., Rose, C.M., Kirkpatrick, D.S., Modrusan, Z., Chen, Y.J., Dugger, D.L., Gandham, V., Xie, L., Newton, K., Dixit, V.M., 2020. Ubiquitin Ligase COP1 Suppresses Neuroinflammation by Degrading c/EBPbeta in Microglia. *Cell.* 182(5), 1156-1169 e1112.
<https://doi.org/10.1016/j.cell.2020.07.011>.
- Negron, A.L., Radovick, S., 2020. High-Fat Diet Alters LH Secretion and Pulse Frequency in Female Mice in an Estrous Cycle-Dependent Manner. *Endocrinology.* 161(10). <https://doi.org/10.1210/endo/bqaa146>.
- Nguyen, P.T., Dorman, L.C., Pan, S., Vainchtein, I.D., Han, R.T., Nakao-Inoue, H., Taloma, S.E., Barron, J.J., Molofsky, A.B., Kheirbek, M.A., Molofsky, A.V., 2020. Microglial Remodeling of the Extracellular Matrix Promotes Synapse Plasticity. *Cell.* 182(2), 388-403 e315. <https://doi.org/10.1016/j.cell.2020.05.050>.
- Nie, X., Kitaoka, S., Tanaka, K., Segi-Nishida, E., Imoto, Y., Ogawa, A., Nakano, F., Tomohiro, A., Nakayama, K., Taniguchi, M., Mimori-Kiyosue, Y., Kakizuka, A., Narumiya, S., Furuyashiki, T., 2018. The Innate Immune Receptors TLR2/4 Mediate Repeated Social Defeat Stress-Induced Social Avoidance through Prefrontal Microglial Activation. *Neuron.* 99(3), 464-479 e467.
<https://doi.org/10.1016/j.neuron.2018.06.035>.
- Nugent, A.A., Lin, K., van Lengerich, B., Lianoglou, S., Przybyla, L., Davis, S.S., Llapashtica, C., Wang, J., Kim, D.J., Xia, D., Lucas, A., Baskaran, S., Haddick, P.C.G., Lenser, M., Earr, T.K., Shi, J., Dugas, J.C., Andreone, B.J., Logan, T., Solanoy, H.O., Chen, H., Srivastava, A., Poda, S.B., Sanchez, P.E., Watts, R.J., Sandmann, T., Astarita, G., Lewcock, J.W., Monroe, K.M., Di Paolo, G., 2020. TREM2 Regulates Microglial Cholesterol Metabolism upon Chronic Phagocytic Challenge. *Neuron.* 105(5), 837-854 e839.
<https://doi.org/10.1016/j.neuron.2019.12.007>.
- O'Neill, L.A., Kishton, R.J., Rathmell, J., 2016. A guide to immunometabolism for immunologists. *Nat Rev Immunol.* 16(9), 553-565.
<https://doi.org/10.1038/nri.2016.70>.

- Orihuela, R., McPherson, C.A., Harry, G.J., 2016. Microglial M1/M2 polarization and metabolic states. *Br J Pharmacol.* 173(4), 649-665.
<https://doi.org/10.1111/bph.13139>.
- Osibogun, O., Ogunmoroti, O., Michos, E.D., 2020. Polycystic ovary syndrome and cardiometabolic risk: Opportunities for cardiovascular disease prevention. *Trends Cardiovasc Med.* 30(7), 399-404. <https://doi.org/10.1016/j.tcm.2019.08.010>.
- Park, S.H., Farooq, M.A., Gaertner, S., Bruckert, C., Qureshi, A.W., Lee, H.H., Benrahla, D., Pollet, B., Stephan, D., Ohlmann, P., Lessinger, J.M., Mayoux, E., Auger, C., Morel, O., Schini-Kerth, V.B., 2020. Empagliflozin improved systolic blood pressure, endothelial dysfunction and heart remodeling in the metabolic syndrome ZSF1 rat. *Cardiovasc Diabetol.* 19(1), 19.
<https://doi.org/10.1186/s12933-020-00997-7>.
- Pasciuto, E., Burton, O.T., Roca, C.P., Lagou, V., Rajan, W.D., Theys, T., Mancuso, R., Tito, R.Y., Kouser, L., Callaerts-Vegh, Z., de la Fuente, A.G., Prezzemolo, T., Mascali, L.G., Brajic, A., Whyte, C.E., Yshii, L., Martinez-Muriana, A., Naughton, M., Young, A., Moudra, A., Lemaitre, P., Poovathingal, S., Raes, J., De Strooper, B., Fitzgerald, D.C., Dooley, J., Liston, A., 2020. Microglia Require CD4 T Cells to Complete the Fetal-to-Adult Transition. *Cell.* 182(3), 625-640 e624.
<https://doi.org/10.1016/j.cell.2020.06.026>.
- Patel, R., Shah, G., 2018. High-fat diet exposure from pre-pubertal age induces polycystic ovary syndrome (PCOS) in rats. *Reproduction.* 155(2), 141-151.
<https://doi.org/10.1530/REP-17-0584>.
- Perkovic, V., Jardine, M.J., Neal, B., Bompoint, S., Heerspink, H.J.L., Charytan, D.M., Edwards, R., Agarwal, R., Bakris, G., Bull, S., Cannon, C.P., Capuano, G., Chu, P.L., de Zeeuw, D., Greene, T., Levin, A., Pollock, C., Wheeler, D.C., Yavin, Y., Zhang, H., Zinman, B., Meininger, G., Brenner, B.M., Mahaffey, K.W., Investigators, C.T., 2019. Canagliflozin and Renal Outcomes in Type 2 Diabetes and Nephropathy. *N Engl J Med.* 380(24), 2295-2306.
<https://doi.org/10.1056/NEJMoa1811744>.
- Plastira, I., Bernhart, E., Goeritzer, M., DeVaney, T., Reicher, H., Hammer, A., Lohberger, B., Wintersperger, A., Zucol, B., Graier, W.F., Kratky, D., Malle, E., Sattler, W., 2017. Lysophosphatidic acid via LPA-receptor 5/protein kinase D-dependent pathways induces a motile and pro-inflammatory microglial

- phenotype. *J Neuroinflammation*. 14(1), 253.
<https://doi.org/10.1186/s12974-017-1024-1>.
- Pruett, J.E., Torres Fernandez, E.D., Everman, S.J., Vinson, R.M., Davenport, K., Logan, M.K., Ye, S.A., Romero, D.G., Yanes Cardozo, L.L., 2021. Impact of SGLT-2 Inhibition on Cardiometabolic Abnormalities in a Rat Model of Polycystic Ovary Syndrome. *Int J Mol Sci*. 22(5). <https://doi.org/10.3390/ijms22052576>.
- Roberts, J.S., Perets, R.A., Sarfert, K.S., Bowman, J.J., Ozark, P.A., Whitworth, G.B., Blythe, S.N., Toporikova, N., 2017. High-fat high-sugar diet induces polycystic ovary syndrome in a rodent model. *Biol Reprod*. 96(3), 551-562.
<https://doi.org/10.1095/biolreprod.116.142786>.
- Rumpler, E., Takacs, S., Gocz, B., Baska, F., Szenci, O., Horvath, A., Ciofi, P., Hrabovszky, E., Skrapits, K., 2020. Kisspeptin Neurons in the Infundibular Nucleus of Ovariectomized Cats and Dogs Exhibit Unique Anatomical and Neurochemical Characteristics. *Front Neurosci*. 14, 598707.
<https://doi.org/10.3389/fnins.2020.598707>.
- Sa-Nguanmoo, P., Tanajak, P., Kerdphoo, S., Jaiwongkam, T., Prachayasakul, W., Chattipakorn, N., Chattipakorn, S.C., 2017. SGLT2-inhibitor and DPP-4 inhibitor improve brain function via attenuating mitochondrial dysfunction, insulin resistance, inflammation, and apoptosis in HFD-induced obese rats. *Toxicol Appl Pharmacol*. 333, 43-50. <https://doi.org/10.1016/j.taap.2017.08.005>.
- Saitgareeva, A.R., Bulygin, K.V., Gareev, I.F., Beylerli, O.A., Akhmadeeva, L.R., 2020. The role of microglia in the development of neurodegeneration. *Neurol Sci*. 41(12), 3609-3615. <https://doi.org/10.1007/s10072-020-04468-5>.
- Salama, A.A., Amine, E.K., Salem, H.A., Abd El Fattah, N.K., 2015. Anti-Inflammatory Dietary Combo in Overweight and Obese Women with Polycystic Ovary Syndrome. *N Am J Med Sci*. 7(7), 310-316.
<https://doi.org/10.4103/1947-2714.161246>.
- Santiago, J.V., Rayaprolu, S., Xiao, H., Seyfried, N.T., Rangaraju, S., 2021. Identification of state-specific proteomic characteristics of microglia-derived exosomes. *Alzheimers Dement*. 17 Suppl 2, e058665.
<https://doi.org/10.1002/alz.058665>.
- Siew, J.J., Chen, H.M., Chen, H.Y., Chen, H.L., Chen, C.M., Soong, B.W., Wu, Y.R., Chang, C.P., Chan, Y.C., Lin, C.H., Liu, F.T., Chern, Y., 2019. Galectin-3 is

- required for the microglia-mediated brain inflammation in a model of Huntington's disease. *Nat Commun.* 10(1), 3473. <https://doi.org/10.1038/s41467-019-11441-0>.
- Skaznik-Wikiel, M.E., Swindle, D.C., Allshouse, A.A., Polotsky, A.J., McManaman, J.L., 2016. High-Fat Diet Causes Subfertility and Compromised Ovarian Function Independent of Obesity in Mice. *Biol Reprod.* 94(5), 108. <https://doi.org/10.1095/biolreprod.115.137414>.
- Song, W.M., Colonna, M., 2018. The identity and function of microglia in neurodegeneration. *Nat Immunol.* 19(10), 1048-1058. <https://doi.org/10.1038/s41590-018-0212-1>.
- Subbarayan, M.S., Joly-Amado, A., Bickford, P.C., Nash, K.R., 2021. CX3CL1/CX3CR1 signaling targets for the treatment of neurodegenerative diseases. *Pharmacol Ther.* 107989. <https://doi.org/10.1016/j.pharmthera.2021.107989>.
- Tan, S., Ignatenko, S., Wagner, F., Dokras, A., Seufert, J., Zwanziger, D., Dunschen, K., Zakaria, M., Huseinovic, N., Basson, C.T., Mahling, P., Fuhrer, D., Hinder, M., 2021. Licogliflozin versus placebo in women with polycystic ovary syndrome: A randomized, double-blind, phase 2 trial. *Diabetes Obes Metab.* 23(11), 2595-2599. <https://doi.org/10.1111/dom.14495>.
- Tejera, D., Mercan, D., Sanchez-Caro, J.M., Hanan, M., Greenberg, D., Soreq, H., Latz, E., Golenbock, D., Heneka, M.T., 2019. Systemic inflammation impairs microglial A β clearance through NLRP3 inflammasome. *EMBO J.* 38(17), e101064. <https://doi.org/10.15252/embj.2018101064>.
- Thaler, J.P., Yi, C.X., Schur, E.A., Guyenet, S.J., Hwang, B.H., Dietrich, M.O., Zhao, X., Sarruf, D.A., Izgur, V., Maravilla, K.R., Nguyen, H.T., Fischer, J.D., Matsen, M.E., Wisse, B.E., Morton, G.J., Horvath, T.L., Baskin, D.G., Tschop, M.H., Schwartz, M.W., 2012. Obesity is associated with hypothalamic injury in rodents and humans. *J Clin Invest.* 122(1), 153-162. <https://doi.org/10.1172/JCI59660>.
- Tortoriello, D.V., McMinn, J., Chua, S.C., 2004. Dietary-induced obesity and hypothalamic infertility in female DBA/2J mice. *Endocrinology.* 145(3), 1238-1247. <https://doi.org/10.1210/en.2003-1406>.
- True, C., Verma, S., Grove, K.L., Smith, M.S., 2013. Cocaine- and amphetamine-regulated transcript is a potent stimulator of GnRH and kisspeptin cells and may contribute to negative energy balance-induced reproductive

- inhibition in females. *Endocrinology*. 154(8), 2821-2832.
<https://doi.org/10.1210/en.2013-1156>.
- Unger, M.S., Marschallinger, J., Kaindl, J., Klein, B., Johnson, M., Khundakar, A.A., Rossner, S., Heneka, M.T., Couillard-Despres, S., Rockenstein, E., Masliah, E., Attems, J., Aigner, L., 2018. Doublecortin expression in CD8+ T-cells and microglia at sites of amyloid-beta plaques: A potential role in shaping plaque pathology? *Alzheimers Dement*. 14(8), 1022-1037.
<https://doi.org/10.1016/j.jalz.2018.02.017>.
- Valdearcos, M., Douglass, J.D., Robblee, M.M., Dorfman, M.D., Stifler, D.R., Bennett, M.L., Gerritse, I., Fasnacht, R., Barres, B.A., Thaler, J.P., Koliwad, S.K., 2018. Microglial Inflammatory Signaling Orchestrates the Hypothalamic Immune Response to Dietary Excess and Mediates Obesity Susceptibility. *Cell Metab*. 27(6), 1356. <https://doi.org/10.1016/j.cmet.2018.04.019>.
- Valdearcos, M., Robblee, M.M., Benjamin, D.I., Nomura, D.K., Xu, A.W., Koliwad, S.K., 2014. Microglia dictate the impact of saturated fat consumption on hypothalamic inflammation and neuronal function. *Cell Rep*. 9(6), 2124-2138.
<https://doi.org/10.1016/j.celrep.2014.11.018>.
- Vergara, D., Nigro, A., Romano, A., De Domenico, S., Damato, M., Franck, J., Coricciati, C., Wistorski, M., Cardon, T., Fournier, I., Quattrini, A., Salzet, M., Furlan, R., Maffia, M., 2019. Distinct Protein Expression Networks are Activated in Microglia Cells after Stimulation with IFN-gamma and IL-4. *Cells*. 8(6).
<https://doi.org/10.3390/cells8060580>.
- Villani, A., Benjaminsen, J., Moritz, C., Henke, K., Hartmann, J., Norlin, N., Richter, K., Schieber, N.L., Franke, T., Schwab, Y., Peri, F., 2019. Clearance by Microglia Depends on Packaging of Phagosomes into a Unique Cellular Compartment. *Dev Cell*. 49(1), 77-88 e77. <https://doi.org/10.1016/j.devcel.2019.02.014>.
- Voet, S., Mc Guire, C., Hagemeyer, N., Martens, A., Schroeder, A., Wieghofer, P., Daems, C., Staszewski, O., Vande Walle, L., Jordao, M.J.C., Sze, M., Vikkula, H.K., Demeestere, D., Van Imschoot, G., Scott, C.L., Hoste, E., Goncalves, A., Guilliams, M., Lippens, S., Libert, C., Vandenbroucke, R.E., Kim, K.W., Jung, S., Callaerts-Vegh, Z., Callaerts, P., de Wit, J., Lamkanfi, M., Prinz, M., van Loo, G., 2018. A20 critically controls microglia activation and inhibits inflammasome-dependent neuroinflammation. *Nat Commun*. 9(1), 2036.

- <https://doi.org/10.1038/s41467-018-04376-5>.
- Voet, S., Srinivasan, S., Lamkanfi, M., van Loo, G., 2019. Inflammasomes in neuroinflammatory and neurodegenerative diseases. *EMBO Mol Med.* 11(6).
<https://doi.org/10.15252/emmm.201810248>.
- Wachholz, S., Esslinger, M., Plumper, J., Manitz, M.P., Juckel, G., Friebe, A., 2016. Microglia activation is associated with IFN-alpha induced depressive-like behavior. *Brain Behav Immun.* 55, 105-113. <https://doi.org/10.1016/j.bbi.2015.09.016>.
- Wakabayashi, Y., Nakada, T., Murata, K., Ohkura, S., Mogi, K., Navarro, V.M., Clifton, D.K., Mori, Y., Tsukamura, H., Maeda, K., Steiner, R.A., Okamura, H., 2010. Neurokinin B and dynorphin A in kisspeptin neurons of the arcuate nucleus participate in generation of periodic oscillation of neural activity driving pulsatile gonadotropin-releasing hormone secretion in the goat. *J Neurosci.* 30(8), 3124-3132. <https://doi.org/10.1523/JNEUROSCI.5848-09.2010>.
- Wake, H., Moorhouse, A.J., Jinno, S., Kohsaka, S., Nabekura, J., 2009. Resting microglia directly monitor the functional state of synapses in vivo and determine the fate of ischemic terminals. *J Neurosci.* 29(13), 3974-3980.
<https://doi.org/10.1523/JNEUROSCI.4363-08.2009>.
- Wang, Z., Liu, D., Wang, F., Liu, S., Zhao, S., Ling, E.A., Hao, A., 2012. Saturated fatty acids activate microglia via Toll-like receptor 4/NF-kappaB signalling. *Br J Nutr.* 107(2), 229-241. <https://doi.org/10.1017/S0007114511002868>.
- Wekker, V., van Dammen, L., Koning, A., Heida, K.Y., Painter, R.C., Limpens, J., Laven, J.S.E., Roeters van Lennep, J.E., Roseboom, T.J., Hoek, A., 2020. Long-term cardiometabolic disease risk in women with PCOS: a systematic review and meta-analysis. *Hum Reprod Update.* 26(6), 942-960.
<https://doi.org/10.1093/humupd/dmaa029>.
- Wen, X., Han, Z., Liu, S.J., Hao, X., Zhang, X.J., Wang, X.Y., Zhou, C.J., Ma, Y.Z., Liang, C.G., 2020. Phycocyanin Improves Reproductive Ability in Obese Female Mice by Restoring Ovary and Oocyte Quality. *Front Cell Dev Biol.* 8, 595373.
<https://doi.org/10.3389/fcell.2020.595373>.
- Wingo, A.P., Fan, W., Duong, D.M., Gerasimov, E.S., Dammer, E.B., Liu, Y., Harerimana, N.V., White, B., Thambisetty, M., Troncoso, J.C., Kim, N., Schneider, J.A., Hajjar, I.M., Lah, J.J., Bennett, D.A., Seyfried, N.T., Levey, A.I., Wingo, T.S., 2020. Shared proteomic effects of cerebral atherosclerosis and Alzheimer's disease

- on the human brain. *Nat Neurosci.* 23(6), 696-700.
<https://doi.org/10.1038/s41593-020-0635-5>.
- Wiviott, S.D., Raz, I., Bonaca, M.P., Mosenzon, O., Kato, E.T., Cahn, A., Silverman, M.G., Zelniker, T.A., Kuder, J.F., Murphy, S.A., Bhatt, D.L., Leiter, L.A., McGuire, D.K., Wilding, J.P.H., Ruff, C.T., Gause-Nilsson, I.A.M., Fredriksson, M., Johansson, P.A., Langkilde, A.M., Sabatine, M.S., Investigators, D.-T., 2019. Dapagliflozin and Cardiovascular Outcomes in Type 2 Diabetes. *N Engl J Med.* 380(4), 347-357. <https://doi.org/10.1056/NEJMoa1812389>.
- Wu, S., Xue, P., Grayson, N., Bland, J.S., Wolfe, A., 2019. Bitter Taste Receptor Ligand Improves Metabolic and Reproductive Functions in a Murine Model of PCOS. *Endocrinology.* 160(1), 143-155. <https://doi.org/10.1210/en.2018-00711>.
- Wu, Y., Li, Y., Liao, X., Wang, Z., Li, R., Zou, S., Jiang, T., Zheng, B., Duan, P., Xiao, J., 2017. Diabetes Induces Abnormal Ovarian Function via Triggering Apoptosis of Granulosa Cells and Suppressing Ovarian Angiogenesis. *Int J Biol Sci.* 13(10), 1297-1308. <https://doi.org/10.7150/ijbs.21172>.
- Yang, J.A., Yasrebi, A., Snyder, M., Roepke, T.A., 2016. The interaction of fasting, caloric restriction, and diet-induced obesity with 17beta-estradiol on the expression of KNDy neuropeptides and their receptors in the female mouse. *Mol Cell Endocrinol.* 437, 35-50. <https://doi.org/10.1016/j.mce.2016.08.008>.
- Yi, C.X., Walter, M., Gao, Y., Pitra, S., Legutko, B., Kalin, S., Layritz, C., Garcia-Caceres, C., Bielohuby, M., Bidlingmaier, M., Woods, S.C., Ghanem, A., Conzelmann, K.K., Stern, J.E., Jastroch, M., Tschop, M.H., 2017. TNFalpha drives mitochondrial stress in POMC neurons in obesity. *Nat Commun.* 8, 15143. <https://doi.org/10.1038/ncomms15143>.
- Zhang, J., Yi, S., Li, Y., Xiao, C., Liu, C., Jiang, W., Yang, C., Zhou, T., 2020. The antidepressant effects of asperosaponin VI are mediated by the suppression of microglial activation and reduction of TLR4/NF-kappaB-induced IDO expression. *Psychopharmacology (Berl).* 237(8), 2531-2545. <https://doi.org/10.1007/s00213-020-05553-5>.
- Zhang, X., Zhang, G., Zhang, H., Karin, M., Bai, H., Cai, D., 2008. Hypothalamic IKKbeta/NF-kappaB and ER stress link overnutrition to energy imbalance and obesity. *Cell.* 135(1), 61-73. <https://doi.org/10.1016/j.cell.2008.07.043>.
- Ziarniak, K., Kolodziejcki, P.A., Pruszyńska-Oszmalek, E., Dudek, M., Kallo, I.,

- Sliwowska, J.H., 2020. Effects of Ovariectomy and Sex Hormone Replacement on Numbers of Kisspeptin-, Neurokinin B- and Dynorphin A-immunoreactive Neurons in the Arcuate Nucleus of the Hypothalamus in Obese and Diabetic Rats. *Neuroscience*. 451, 184-196. <https://doi.org/10.1016/j.neuroscience.2020.10.003>.
- Ziarniak, K., Kolodziejcki, P.A., Pruszyńska-Oszmalek, E., Kallomicron, I., Sliwowska, J.H., 2018. High-fat diet and type 2 diabetes induced disruption of the oestrous cycle and alteration of hormonal profiles, but did not affect subpopulations of KNDy neurones in female rats. *J Neuroendocrinol*. 30(11), e12651. <https://doi.org/10.1111/jne.12651>.
- Zinman, B., Wanner, C., Lachin, J.M., Fitchett, D., Bluhmki, E., Hantel, S., Mattheus, M., Devins, T., Johansen, O.E., Woerle, H.J., Broedl, U.C., Inzucchi, S.E., Investigators, E.-R.O., 2015. Empagliflozin, Cardiovascular Outcomes, and Mortality in Type 2 Diabetes. *N Engl J Med*. 373(22), 2117-2128. <https://doi.org/10.1056/NEJMoA1504720>.
- Zusso, M., Lunardi, V., Franceschini, D., Pagetta, A., Lo, R., Stifani, S., Frigo, A.C., Giusti, P., Moro, S., 2019. Ciprofloxacin and levofloxacin attenuate microglia inflammatory response via TLR4/NF-κB pathway. *J Neuroinflammation*. 16(1), 148. <https://doi.org/10.1186/s12974-019-1538-9>.

All changes were made using colored texts for easy detection in the revision.

Referee #1:

Chen et al. (JP-TR-2022-283259) provide a review about the effects of SGLT2 inhibitors on high-fat diet-induced hypothalamic-pituitary-ovarian axis disorders. The topic is timely and of general interest for physiologists, pathophysiologists and researchers in other disciplines. The manuscript partially lacks clarity and scientific depth, however. The authors might consider the following criticism to improve clarity and to strengthen the conclusions.

Major points:

Question 1. Page 8 (PDF). Sentence, 'Thus, HFD causes HPO axis dysfunction traversing multiple levels, from the central regulating neurons down to the peripheral ovarian glands.' Dysfunction in 'central regulating neurons' is not that obvious in the main text. Please, provide more details and/or rephrase.

Response 1. We have added contents on the central neurons responsible for ovulation.

Question 2. Page 10, Figure 2 (legend) and elsewhere. Sentence, 'The inflammatory cytokines include tumor necrosis factor- α (TNF- α), interleukin-1 β (IL-1 β), interleukin-18 (IL-18), superoxide, nitric oxide (NO), reactive oxygen species (ROS), and proteases.' ROS and RNS are not cytokines. What is the difference between 'superoxide' and 'ROS'? Please, check and rephrase.

Response 2. We have organized and rewritten the information.

Question 3. Page 11, Figure 2 and elsewhere. The terminology of 'resting' microglia is obsolete. Please, use 'surveying' or 'homeostatic' microglia.

Response 3. We have used "homeostatic microglia".

Question 4. Page 11 and Figure 2. Please, differentiate between general markers and activation markers of microglia.

Response 4. We have revised this section.

Question 5. Page 11. The observation (concept) of 'over-reactive microglia' needs to be discussed in more detail (experimental settings and models, stimuli/triggers etc.).

Response 5. We have discussed in detail according to original references.

Question 6. Page 14 and Figure 3. Sentence, 'Previous studies suggest that a Toll-like receptor 4 (TLR4)/ nuclear factor kappa-light-chain-enhancer of activated B cells (NF- κ B) signaling pathway in microglia injury (Muhammad et al., 2019).'

a) Is 'microglia-mediated injury' meant? Please, check and rephrase.

Response 6a. Previous studies have suggested the role of the Toll-like receptor 4 (TLR4)/ nuclear factor kappa B (NF- κ B) signaling pathway in microglial injury.

b) Activation of TLR-4 in microglia is indeed proinflammatory. However, many studies have shown that sole TLR-4 activation in microglia is not sufficient to induce neuronal death (e.g., Chao et al., J Immunol, 1992; Duport and Garthwaite, Neuroscience, 2005; Papageorgiou et al., Proc Natl Acad Sci U S A, 2016; Schilling et al., Brain Behav Immun, 2021). Please, discuss both context and effects of TLR-4 activation in microglia in more detail.

Response 6b. We have reviewed TLRs and microglia activation.

Question 7. Pages 19 ff and Figure 4.

a) Please, discuss the role of microglia in 'hypothalamic inflammation' in more detail.

Response 7a. We have reviewed the possible pathophysiological mechanisms of HFD-induced microglia activation in hypothalamic inflammation in part 2. We also added data about the role of microglia in hypothalamic inflammation.

b) Please, include microglia and its inflammatory release products in Figure 4.

Response 7b. We have added microglia and its inflammatory release products in Figure 4.

Question 8. Page 21 and elsewhere. Please, introduce the features of the drug empagliflozin. This also applies to other drugs discussed.

Response 8. We have briefly described the features of drugs.

Question 9. Pages 22 ff and Figure 5. It is not really clear how SGLT2 inhibitors might attenuate activation of microglia that might proliferate and become glycolytic (e.g., Gosh et al., *Glia*, 2018; Bernier et al., *Trends Neurosci*, 2020). Please, discuss this issue in more detail.

Response 9. We have discussed this issue about the effect of SGLT2 inhibitors on microglia metabolism and amelioration of high-fat-induced hypothalamic inflammation

Question 10. Figure 5. Please, make clear which pathways relate to microglia and which not.

Response 10. Figure 5 has been revised.

Minor points:

Question 11. Abbreviations. Please, generally reduce the use of abbreviations for the sake of readability (e.g., GS, ARC, NKB, CA etc.). Please, also reduce the use of uncommon and less intuitive abbreviations (e.g., SGLT2i, VO etc.).

Response 11. We have reduced these abbreviations. inhibitor

Question 12. Page 8. What are 'master neurons'? Please, rephrase.

Response 12. We have revised "Ovulation is triggered by LH surge, which is secondary to the surge-mode release of GnRH by the master neurons, i.e. GnRH neurons."

Question 13. Page 8. Sentence, '(...) high levels of pro-inflammatory cytokines and increased macrophage infiltration (...)' Please, specify location (tissue and/or layer).

Response 13. We have added "in the ovary"

Question 14. Page 13. Sentence, 'Microglial inflammatory activation in the medio-basal hypothalamus (MBH) following HFD-feeding is associated with (...)'. Please, specify 'neuronal functional changes'.

Response 14. According to reviewer's comments, we have revised the content and added primary article citations.

Question 15. Page 13. Sentence, 'Of interest, Tschop et al. have observed that microglial cells exert their function in a strict circadian pattern, (...)' This is somewhat misleading and likely not true for acute infections etc. Please, rephrase.

Response 15. We have rephrased this sentence after reading the primary article.

Question 16. Figure 4 and legend. How relevant is 'neurogenesis' in the adult hypothalamus? Please, comment on.

Response 16: We have added comment and primary article citations on neurogenesis. **(green letter)**

Referee #2:

The manuscript "SGLT2 inhibitor ameliorates high-fat diet-induced hypothalamic-pituitary-ovarian axis disorders" envisages role of SGLT2 inhibitors in various disorders (cognitive, PCOD etc) and their "hypothetical" mechanism of action in these conditions, which is attributed to the reduction in brain oxidative stress and the amelioration of hypothalamic neuroinflammation. The authors have build their review on following major points:

1. Consumption of HFD causes hypothalamic-pituitary-ovarian (HPO) axis dysfunction
2. HFD-induced hypothalamic inflammation is involved in HPO axis dysfunction
3. SGLT2i improve female reproductive function
4. The possible mechanism of SGLT2 inhibitor ameliorates HPO axis dysfunction

Overall the manuscript has been written meticulously, with up to date review of literature. Hypothesis put forward has some basis, which can be deliberated upon. However, the manuscript needs proper language editing, as many statements/ points lead to confusion because of language problems. I have highlighted some points in the manuscript that need to be addressed (file attached for reference). Some of the important points that need to be addressed are:

1. "Both obese mice and non-obese mice were subfertility and show impaired ovarian function regardless of the obese phenotype"
2. "Upon activation, NLRP3 interaction with ASC through CARD-CARD result in recruiting pro-caspase 1, which then pro-caspase 1 is converted into caspase 1".
3. "TLR4 loss-of-function mutation or pharmacological inhibition of TLR4 alleviates HFD-induced weight gain and attenuates HFD induced hypothalamic highly expression of inflammation factors such as IL-6 and IL-10"
4. "Yang et al. revealed HFD-fed was no effects in the arcuate Tac2 mRNA (which encodes NKB) expression in female mice".
5. " This somehow, provide hints to form an HFD-hypothalamic inflammation"
6. "significant decreased body weight gain"
7. " Black solid arrow, stimulate; red blunt line, inhibit; green solid arrow, stimulating effect of SGLT2i; green blunt line, inhibitory effect of SGLT2i".

Responses:

All points highlighted by reviewer have been corrected and carefully organized **(green letter)**. Thank you very much for your careful reading.

Dear Professor Chen,

Re: JP-TR-2022-283259R1 "SGLT2 inhibitor ameliorates high-fat diet-induced hypothalamic-pituitary-ovarian axis disorders" by Xiaolin Chen, Lili Huang, Ling Cui, Zhuoni Xiao, Xiaoxing Xiong, and Chen Chen

Thank you for submitting your Topical Review to The Journal of Physiology. It has been assessed by a Reviewing Editor and by 2 expert referees and I am pleased to tell you that it is considered to be acceptable for publication following satisfactory revision.

The reports are copied at the end of this email. Please address all of the points and incorporate all requested revisions, or explain in your Response to Referees why a change has not been made.

NEW POLICY: In order to improve the transparency of its peer review process The Journal of Physiology publishes online as supporting information the peer review history of all articles accepted for publication. Readers will have access to decision letters, including all Editors' comments and referee reports, for each version of the manuscript and any author responses to peer review comments. Referees can decide whether or not they wish to be named on the peer review history document.

I hope you will find the comments helpful and have no difficulty in revising your manuscript within 4 weeks.

Your revised manuscript should be submitted online using the links in Author Tasks Link Not Available. This link is to the Corresponding Author's own account, if this will cause any problems when submitting the revised version please contact us.

You should upload:

- A Word file of the complete text (including any Tables);
- An Abstract Figure, (with accompanying Legend in the article file)
- Each figure as a separate, high quality, file;
- A full Response to Referees;
- A copy of the manuscript with the changes highlighted.
- Author profile. A short biography (no more than 100 words for one author or 150 words in total for two authors) and a portrait photograph of the two leading authors on the paper. These should be uploaded, clearly labelled, with the manuscript submission. Any standard image format for the photograph is acceptable, but the resolution should be at least 300 dpi and preferably more.

- A 'Cover Art' file for consideration as the Issue's cover image;
- Appropriate Supporting Information (Video, audio or data set https://jp.msubmit.net/cgi-bin/main.plex?form_type=display_requirements#supp).

To create your 'Response to Referees' copy all the reports, including any comments from the Senior and Reviewing Editors into a Word, or similar, file and respond to each point in colour or CAPITALS. Upload this when you submit your revision.

I look forward to receiving your revised submission.

Yours sincerely,

Professor Laura Bennet
Senior Editor
The Journal of Physiology
<https://jp.msubmit.net>
<http://jp.physoc.org>
The Physiological Society
Hodgkin Huxley House
30 Farringdon Lane
London, EC1R 3AW
UK
<http://www.physoc.org>
<http://journals.physoc.org>

EDITOR COMMENTS

Both referees recommended acceptance, however, Referee 2 would like a small change to be made: "Although it is still unclear what exact mechanism mediates the interaction between HFD and KNDy neurons, it supports an inflammatory

theory to some extent. How? (basis of your opinion) and to what extent? Please explain

page 5: Both obese mice and non-obese mice were subfertility and had impaired ovarian function regardless of the obese phenotype. is it subfertility or in a state of subfertility."

Please see also 'Required Items' below.

REFeree COMMENTS

Referee #1:

The authors addressed my points and made appropriate changes in text and figures.

Referee #2:

My concerns have been taken care of, however still at one place (Page 18) there is an ambiguous statement "Although it is still unclear what exact mechanism mediates the interaction between HFD and KNDy neurons, it supports an inflammatory theory to some extent. How?(basis of your opinion) and to what extent?Please explain

page 5: Both obese mice and non-obese mice were subfertility and had impaired ovarian function regardless of the obese phenotype. is it subfertility or in a state of subfertility.

REQUIRED ITEMS:

-Your MS must include a complete "Additional information section" with the following 4 headings and content:

Competing Interests: A statement regarding competing interests. If there are no competing interests, a statement to this effect must be included. All authors should disclose any conflict of interest in accordance with journal policy.

Author contributions: Each author should take responsibility for a particular section of the study and have contributed to writing the paper. Acquisition of funding, administrative support or the collection of data alone does not justify authorship; these contributions to the study should be listed in the Acknowledgements. Additional information such as 'X and Y have contributed equally to this work' may be added as a footnote on the title page.

It must be stated that all authors approved the final version of the manuscript and that all persons designated as authors qualify for authorship, and all those who qualify for authorship are listed.

Funding: Authors must indicate all sources of funding, including grant numbers. If authors have not received funding, this must be stated.

It is the responsibility of authors funded by RCUK to adhere to their policy regarding funding sources and underlying research material. The policy requires funding information to be included within the acknowledgement section of a paper. Guidance on how to acknowledge funding information is provided by the Research Information Network. The policy also requires all research papers, if applicable, to include a statement on how any underlying research materials, such as data, samples or models, can be accessed. However, the policy does not require that the data must be made open. If there are considered to be good or compelling reasons to protect access to the data, for example commercial confidentiality or legitimate sensitivities around data derived from potentially identifiable human participants, these should be included in the statement.

Acknowledgements: Acknowledgements should be the minimum consistent with courtesy. The wording of acknowledgements of scientific assistance or advice must have been seen and approved by the persons concerned. This section should not include details of funding.

END OF COMMENTS

1st Confidential Review

25-Jul-2022

Editor-in-Chief

J Physiology

Re: JP-TR-2022-283259R1

Title: "SGLT2 inhibitor ameliorates high-fat diet-induced hypothalamic-pituitary-ovarian axis disorders"

Dear Peter,

Thank you very much for considering our review for publication in your Journal. We have carefully studied the comments of the Referee 2 and made this revision accordingly.

We are look forward to hearing from you soon.

With kind regards,

Sincerely yours,

Chen et al.

Point-by-point responses to reviewers' comments:

First of all, we thank the Reviewers for their positive and constructive comments and suggestions.

REFEREE COMMENTS:

Referee #1:

The authors addressed my points and made appropriate changes in text and figures.

Response:

Thank you for your approval.

Referee #2:

My concerns have been taken care of, however still at one place (Page 18) there is an ambiguous statement "Although it is still unclear what exact mechanism mediates the interaction between HFD and KNDy neurons, it supports an inflammatory theory to some extent. How?(basis of your opinion) and to what extent?Please explain

Response:

After careful reading of the literature and context, we have altered the sentence to "Although it is still unclear what exact mechanism mediates the interaction between HFD and KNDy neurons, indirect evidence supports the inflammatory theory to some extent."

page 5: Both obese mice and non-obese mice were subfertility and had impaired ovarian function regardless of the obese phenotype. is it subfertility or in a state of subfertility.

Response:

Both obese mice and non-obese mice were in a state of subfertility and had impaired ovarian function regardless of the obese phenotype.

Dear Professor Chen,

Re: JP-TR-2022-283259R2 "SGLT2 inhibitor ameliorates high-fat diet-induced hypothalamic-pituitary-ovarian axis disorders" by Xiaolin Chen, Lili Huang, Ling Cui, Zhuoni Xiao, Xiaoxing Xiong, and Chen Chen

I am pleased to tell you that your Topical Review article has been accepted for publication in The Journal of Physiology, subject to any modifications to the text that may be required by the Journal Office to conform to House rules.

NEW POLICY: In order to improve the transparency of its peer review process The Journal of Physiology publishes online as supporting information the peer review history of all articles accepted for publication. Readers will have access to decision letters, including all Editors' comments and referee reports, for each version of the manuscript and any author responses to peer review comments. Referees can decide whether or not they wish to be named on the peer review history document.

The last Word version of the paper submitted will be used by the Production Editors to prepare your proof. When this is ready you will receive an email containing a link to Wiley's Online Proofing System. The proof should be checked and corrected as quickly as possible.

All queries at proof stage should be sent to tjp@wiley.com.

The accepted version of the manuscript will be published online, prior to copy editing in the Accepted Articles section.

Are you on Twitter? Once your paper is online, why not share your achievement with your followers. Please tag The Journal (@jphysiol) in any tweets and we will share your accepted paper with our 22,000+ followers!

Yours sincerely,

Professor Laura Bennet
Senior Editor
The Journal of Physiology
<https://jp.msubmit.net>
<http://jp.physoc.org>
The Physiological Society
Hodgkin Huxley House
30 Farringdon Lane
London, EC1R 3AW
UK
<http://www.physoc.org>
<http://journals.physoc.org>

*** IMPORTANT NOTICE ABOUT OPEN ACCESS ***

To assist authors whose funding agencies mandate public access to published research findings sooner than 12 months after publication The Journal of Physiology allows authors to pay an open access (OA) fee to have their papers made freely available immediately on publication.

You will receive an email from Wiley with details on how to register or log-in to Wiley Authors Services where you will be able to place an OnlineOpen order.

You can check if you funder or institution has a Wiley Open Access Account here <https://authorservices.wiley.com/author-resources/Journal-Authors/licensing-and-open-access/open-access/author-compliance-tool.html>

Your article will be made Open Access upon publication, or as soon as payment is received.

If you wish to put your paper on an OA website such as PMC or UKPMC or your institutional repository within 12 months of publication you must pay the open access fee, which covers the cost of publication.

OnlineOpen articles are deposited in PubMed Central (PMC) and PMC mirror sites. Authors of OnlineOpen articles are permitted to post the final, published PDF of their article on a website, institutional repository, or other free public server, immediately on publication.

Note to NIH-funded authors: The Journal of Physiology is published on PMC 12 months after publication, NIH-funded authors DO NOT NEED to pay to publish and DO NOT NEED to post their accepted papers on PMC.

EDITOR COMMENTS

Reviewing Editor:

Thank you for making these changes and the manuscript is now approved.

Senior Editor:

Thank you for your interesting review.

2nd Confidential Review

14-Aug-2022